# mL-BFGS: A Momentum-based L-BFGS for Distributed Large-Scale Neural Network Optimization

**Yue Niu**  *yueniu@usc.edu*
*Department of Electrical and Computer Engineering*
*University of Southern California*

**Zalan Fabian**  *zfabian@usc.edu*
*Department of Electrical and Computer Engineering*
*University of Southern California*

**Sunwoo Lee**  *sunwool@inha.ac.kr*
*Department of Computer Science and Engineering,*
*Inha University*

**Mahdi Soltanolkotabi**  *soltanol@usc.edu*
*Department of Electrical and Computer Engineering*
*University of Southern California*

**Salman Avestimehr**  *avestime@usc.edu*
*Department of Electrical and Computer Engineering*
*University of Southern California*

**Reviewed on OpenReview:** *https://openreview.net/forum?id=9jnsPp8DP3*

## Abstract

Quasi-Newton methods still face significant challenges in training large-scale neural networks due to additional compute costs in the Hessian related computations and instability issues in stochastic training. A well-known method, L-BFGS that efficiently approximates the Hessian using history parameter and gradient changes, suffers convergence instability in stochastic training. So far, attempts that adapt L-BFGS to large-scale stochastic training incur considerable extra overhead, which offsets its convergence benefits in wall-clock time. In this paper, we propose mL-BFGS, a lightweight momentum-based L-BFGS algorithm that paves the way for quasi-Newton (QN) methods in large-scale distributed deep neural network (DNN) optimization. mL-BFGS introduces a nearly cost-free momentum scheme into L-BFGS update and greatly reduces stochastic noise in the Hessian, therefore stabilizing convergence during stochastic optimization. For model training at a large scale, mL-BFGS approximates a block-wise Hessian, thus enabling distributing compute and memory costs across all computing nodes. We provide a supporting convergence analysis for mL-BFGS in stochastic settings. To investigate mL-BFGS's potential in large-scale DNN training, we train benchmark neural models using mL-BFGS and compare performance with baselines (SGD, Adam, and other quasi-Newton methods). Results show that mL-BFGS achieves both noticeable iteration-wise and wall-clock speedup.

## 1 Introduction

In supervised learning, a typical task is to minimize a empirical risk function,

$$\min_{\boldsymbol{\theta}} \mathcal{L}(\boldsymbol{\theta}; \mathcal{X}) := \frac{1}{N} \sum_{i=1}^{N} \ell(\boldsymbol{\theta}; x_i, y_i), \qquad (1)$$

where $\boldsymbol{\theta} \in \mathbb{R}^d$ denote the parameters to be optimized, and $\mathcal{X}$ represents training samples $\{x_i, y_i\}_{i=1}^{N}$.

At present, stochastic gradient descent method (SGD) and its variants, such as Adam (Kingma & Ba, 2014) are the preferred methods to optimize parameters $\boldsymbol{\theta}$ due to their simplicity, especially for large-scale machine

learning problems. Nevertheless, second-order or quasi-Newton (QN) methods have also been extensively investigated due to their superior convergence over gradient descent (GD) in strongly convex optimization settings (Gao & Goldfarb, 2019; Rodomanov & Nesterov, 2021).

However, the Achilles heel of QN methods that is impeding their wide adoption for large-scale machine learning problems is their substantial compute and memory costs. Specifically, these barriers stem from attaining second-order information, including computing and storing the Hessian, performing matrix inversion, etc. For large-scale neural networks, these operations pose daunting challenges to QN's implementations and significantly affect running time. Moreover, second-order methods are inherently hard to parallelize as the Hessian inversion usually involves many sequential steps, making it difficult to leverage large-scale distributed systems to partition computations and memory across multiple nodes. As a result, even though superior convergence performance is observed in strongly convex settings, it still remains unclear how to convey such benefits to large-scale model training in distributed settings.

Due to the prohibitive challenge above, approximation methods are getting increasing attention that attain second-order information by formulating the Hessian inverse in different ways. These methods, to some extent, open the door for second-order methods to large-scale machine learning. Among them, two lines have proved very promising. The first one arises from the Fisher information matrix $I$ (expectation of the Hessian under a negative log-likelihood loss). Methods such as KFAC (Martens & Grosse, 2015; Ba et al., 2017; Pauloski et al., 2020) first approximate $I$ and then simplify matrix inversion by decomposing $I$ into small submatrices. However, the considerable overhead for obtaining the empirical Fisher information and its inverse greatly neutralizes its faster per-iteration convergence promises. Another line of the approximation methods directly approximates the Hessian inverse via BFGS update. With pairs of history gradient and parameter changes, BFGS directly approaches the Hessian inverse with no additional costs on computing and storing the Hessian matrix. However, BFGS or its variant L-BFGS (Nocedal, 1980) has not proved efficient in large-scale stochastic optimization, where convergence instability is commonly observed. Additional operations introduced in recent works stabilize the training but with substantial costs (Mokhtari & Ribeiro, 2015; Moritz et al., 2016; Gower et al., 2016). For instance, Moritz et al. (2016) uses a separate large batch of inputs to compute the consistent gradient and parameter changes, which inevitably increase the wall-clock time. Moreover, moving to distributed systems, these methods are not amenable to efficient parallelization due to either direct matrix inverse (e.g., KFAC) or the iterative procedure in BFGS-like methods.

To simultaneously mitigate the challenges of compute and memory costs, and scalability in distributed systems, we propose mL-BFGS, a stable distributed BFGS variant that preserves the convergence promises of second-order methods, with modest compute and memory costs compared to other QN methods. mL-BFGS addresses the aforementioned barriers as follows. First, mL-BFGS introduces an almost cost-free momentum scheme into the BFGS update rule to stabilize the optimization. By using the momentum of the *history statistics* (parameter and gradient changes) to estimate the Hessian inverse, mL-BFGS smooths out the approximation with no needs of costly variance reduction methods (e.g., a separate large batch size to estimate the Hessian). Hence, both stability and compute efficiency are achieved. For efficient parallelization and low memory footprint in distributed systems, mL-BFGS presents a more generic block-wise Hessian approximation with each block consisting of one or multiple layers. During optimization, each node only computes and stores statistics for one block rather than the full Hessian. As a result, mL-BFGS can perform the Hessian inverse and gradient conditioning in a distributed way with marginal communication and synchronization overhead.

Our theoretical analysis shows mL-BFGS effectively suppresses noise in the Hessian approximation and achieves stable convergence. Empirical evaluations show that, on benchmark datasets, CIFAR-10 and ImageNet, and models such as ResNet and Vision Transformer, mL-BFGS achieves a faster per-iteration convergence compared to SGD and Adam. Furthermore, due to the lightweight momentum-based Hessian approximation, mL-BFGS needs a much shorter wall-clock time to achieve the target accuracy compared to SGD, Adam and other QN methods such as KFAC.

In summary, our main contributions are as follows:

1. We develop mL-BFGS, a distributed stochastic QN method for large-scale models, that achieves fast and stable convergence with low computational complexity.

2. We provide theoretical analyses that show mL-BFGS significantly mitigates adverse effects of stochastic noise on the Hessian approximation and achieves stable convergence for stochastic optimization problems.

3. We provide complexity analyses that demonstrate mL-BFGS incurs much less complexity than other QN methods, leading to reductions in overall wall-clock training time.

4. Finally, we carry out comprehensive evaluations on various models and datasets that show mL-BFGS empirically delivers faster per-iteration and wall-clock convergence compared to SGD, Adam, and other second-order optimizers.

## 2 Preliminaries

The risk function $\mathcal{L}(\boldsymbol{\theta}, \mathcal{X})$ in Eq (1) is usually optimized through a form of gradient descent as:

$$\boldsymbol{\theta}_{t+1} = \boldsymbol{\theta}_t - \eta_t \cdot \hat{H} \cdot \boldsymbol{g}_t, \tag{2}$$

where $\eta_t$ denotes step size (learning rate) at iteration $t$ and $\hat{H}$ is a gradient pre-conditioner. In stochastic training, gradients are evaluated on a mini-batch input $\mathbf{X}_t \subseteq \mathcal{X}$, namely $\boldsymbol{g}_t = \nabla_{\boldsymbol{\theta}} \mathcal{L}(\boldsymbol{\theta}_t, \mathbf{X}_t)$.

If $\hat{H}$ is an identity matrix, the update above is reduced to SGD, whereas if $\hat{H}$ is a diagonal matrix, it becomes an adaptive training algorithm such as Adagrad (Duchi et al., 2011) or Adam (Kingma & Ba, 2014). To further improve convergence performance, esp. in an ill-conditioned problem (Nesterov, 2003), it is desired to incorporate more second-order information into $\hat{H}$ as done in quasi-Newton (QN) methods.

A prime challenge in QN methods is the evaluation of $\hat{H}$ and in particular its inverse. A well-known Broyden–Fletcher–Goldfarb–Shanno (BFGS) algorithm (Fletcher, 2013) addresses the challenge by formulating the Hessian inverse as a minimization problem:

$$\min_{\hat{H}} \quad \left\| \hat{H} - \hat{H}_{k-1} \right\|^2,$$
$$\text{s.t.} \quad \hat{H} \cdot \boldsymbol{y}_k = \boldsymbol{s}_k, \quad \hat{H} \text{ is symmetric}, \tag{3}$$

where $\boldsymbol{s}_k = \boldsymbol{\theta}_k - \boldsymbol{\theta}_{k-1}$ denotes the parameter changes, and $\boldsymbol{y}_k = \boldsymbol{g}_k - \boldsymbol{g}_{k-1}$ the gradient changes in two consecutive updates [1]. By imposing the *secant* condition during minimization, BFGS gradually attains the curvature information close to the real Hessian. Rodomanov & Nesterov (2021) establishes that BFGS converges to the real Hessian by a greedy strategy of choosing $(\boldsymbol{s}_k, \boldsymbol{y}_k)$.

Knowing $\hat{H}_{k-1}$, the current $\hat{H}$ is obtained via:

$$\hat{H}_k = (I - \rho_k \boldsymbol{y}_k \boldsymbol{s}_k^T)^T \hat{H}_{k-1} (I - \rho_k \boldsymbol{y}_k \boldsymbol{s}_k^T) + \rho_k \boldsymbol{s}_k \boldsymbol{s}_k^T, \tag{4}$$

where $\rho_k = \frac{1}{\boldsymbol{y}_k^T \boldsymbol{s}_k}$. Hence the Hessian inverse $\hat{H}$ is constructed in an iterative manner with no need to compute the Hessian matrix.

To simplify computation in the Hessian-vector product, $\hat{H}$ in BFGS is stored in the form of a sequence of history vectors $\{\boldsymbol{y}_i\}$ and $\{\boldsymbol{s}_i\}$. The matrix-vector product $\hat{H}_k \cdot \boldsymbol{g}_t$ is replaced by a sequence of fast vector-vector products as shown in Algorithm 2 (See Appendix A.1). Furthermore, a limited-memory BFGS, L-BFGS (Nocedal, 1980) is usually adopted that only uses several latest history vectors when approximating the Hessian inverse.

## 3 mL-BFGS

mL-BFGS consists of two crucial techniques to improve convergence stability and scalability of using L-BFGS in large-scale models. First, mL-BFGS introduces momentum into the Hessian approximation. The momentum-based design effectively reduces the adverse effects of stochastic noise while without resorting to costly noise-reduction techniques as in current solutions (Moritz et al., 2016). Second, in distributed training,

---

[1] $k$ rather than $t$ is used in the equation as parameter/gradient used might be different from the one in Eq (2)

mL-BFGS allows a block-wise approximation along arbitrary diagonal blocks. mL-BFGS can flexibly assign each computing node a block that comprises multiple layers to distribute the workload. We describe the method in detail below.

### 3.1 Momentum-based Hessian: Reduce Effects of Stochastic Noise

While momentum is widely used in first-order methods, it is rarely explored in the second-order domain. Surprisingly, we find that momentum is also a crucial component for a stable Hessian approximation. Furthermore, compared to other noise-reduction methods, the momentum-based design is almost cost-free.

In this paper, we apply momentum to past parameters and gradients as

$$
\begin{aligned}
\boldsymbol{\theta} &: \mathcal{M}_{\boldsymbol{\theta}_t} = \beta \cdot \mathcal{M}_{\boldsymbol{\theta}_{t-1}} + (1-\beta)\boldsymbol{\theta}_t, \\
\boldsymbol{g} &: \mathcal{M}_{\boldsymbol{g}_t} = \beta \cdot \mathcal{M}_{\boldsymbol{g}_{t-1}} + (1-\beta)\boldsymbol{g}_t,
\end{aligned}
\tag{5}
$$

where $\boldsymbol{\theta}_t, \boldsymbol{g}_t$ denotes parameters and gradients at $t$-th iteration, $\beta$ is the momentum coefficient.

Following the BFGS update rule in Eq (3) and assuming that $\hat{H}$ is updated for every $T$ mini-batch iterations, $\boldsymbol{s}_k$ and $\boldsymbol{g}_k$ are obtained as

$$
\boldsymbol{s}_k = \mathcal{M}_{\boldsymbol{\theta}_{(k+1)T}} - \mathcal{M}_{\boldsymbol{\theta}_{kT}}, \quad \boldsymbol{y}_k = \mathcal{M}_{\boldsymbol{g}_{(k+1)T}} - \mathcal{M}_{\boldsymbol{g}_{kT}}.
\tag{6}
$$

This simple but effective technique works surprisingly well when gradients are noisy. To intuitively show improvements of using momentum over the vanilla L-BFGS, we visualize the stochastic optimization of a simple quadratic loss function, $\mathcal{L} = \frac{1}{2}\|\boldsymbol{\theta}\|^2$. We simulate gradients in stochastic settings at iteration $t$ as $\boldsymbol{g}_t = \boldsymbol{\theta}_t + \boldsymbol{n}_t$, where $\boldsymbol{n}_t$ denotes the *stochastic noise*. We model the noise as i.i.d. Gaussian, namely $\boldsymbol{n}_t \sim \mathcal{N}(0, \sigma^2)$. Figure 1 shows optimization trajectories for SGD, vanilla L-BFGS, L-BFGS with momentum and the exact 2nd-order method given $\sigma = 0.2$. With the same initial point, we observe that L-BFGS with momentum is as fast as the exact 2nd-order method, and much faster than SGD. On the other hand, vanilla L-BFGS obviously suffers convergence issues due to noisy $\boldsymbol{y}_k$.

**Analysis -** We use a quadratic function as a showcase to analyze important theoretical properties of the momentum-based design:

$$
\mathcal{L}(\boldsymbol{\theta}) = \mathcal{L}(\boldsymbol{\theta}_0) + \boldsymbol{g}(\boldsymbol{\theta}_0)(\boldsymbol{\theta} - \boldsymbol{\theta}_0) + \frac{1}{2}(\boldsymbol{\theta} - \boldsymbol{\theta}_0)^* B(\boldsymbol{\theta} - \boldsymbol{\theta}_0).
$$

The following lemmas and theorem establish theoretical improvement of using momentum over vanilla L-BFGS. Specifically, Lemma 1 shows that momentum in Eq (5) significantly reduces stochastic noise in gradient changes $\boldsymbol{y}_k$. Lemma 2 further provides a theoretical guarantee that using momentum can still obtain the same Hessian approximation as the vanilla L-BFGS algorithm in the noise-free case.

Finally, Theorem 1 shows that in stochastic settings, the achievable loss by L-BFGS is lower bounded by a value depending on noise variance in $\boldsymbol{y}_k$. Combined with Lemma 1 and 2, it is obvious that the momentum-based approximation achieves a much lower bound compared to the vanilla version. Such a conclusion is aligned with our empirical observations in Figure 1.

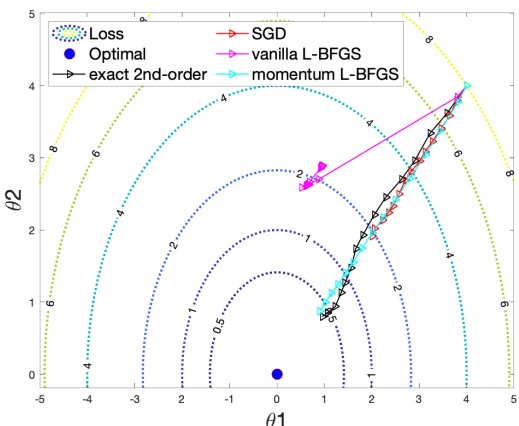

Figure 1: Optimization using SGD, vanilla L-BFGS, L-BFGS with momentum ($\beta = 0.9$) and exact 2nd-order method. Vanilla L-BFGS fails to find the desirable optimization path. However, momentum reduces noise in gradients and significantly stabilizes the optimization.

**Lemma 1.** *Given $\mathcal{L}(\boldsymbol{\theta})$, and assuming $\boldsymbol{g}_t$ in iteration $t$ contains Gaussian noise $\boldsymbol{n}_t$ with zero mean and $E[\|\boldsymbol{n}_t\|^2] \leq \epsilon^2$, then the noise variance in $\mathcal{M}_{\boldsymbol{g}_t}$ in Eq (5) is reduced to $\frac{1-\beta}{1+\beta}\epsilon^2$ as $t \to \infty$. Furthermore, the noise variance in $\boldsymbol{y}_k$ is reduced to $\frac{4(1-\beta)}{1+\beta}\epsilon^2$*

**Lemma 2.** *Considering $\mathcal{L}(\boldsymbol{\theta})$ above, with no noise involved, Eq (6) obtains the same Hessian approximation as the vanilla L-BFGS.*

**Theorem 1.** *Considering $\mathcal{L}(\boldsymbol{\theta})$ with $\lambda I \preceq B \preceq \Lambda I$ where $\Lambda \geq \lambda > 0$, and noise variance in $\boldsymbol{y}_k$ is bounded by $2\epsilon^2$. Assuming during the Hessian estimation, we always choose $\boldsymbol{s}_k, \boldsymbol{y}_k$ such that $\langle \boldsymbol{s}_k, \boldsymbol{y}_k \rangle \geq \alpha \cdot \epsilon \|\boldsymbol{s}_k\|$ with $\alpha > 2$, then $\mathcal{L}(\boldsymbol{\theta}_t) \geq \frac{(\alpha-2)^2 \lambda}{2\Lambda^2} \epsilon^2$ at any iteration $t$.*

*Remark 1.* $\langle \boldsymbol{s}_k, \boldsymbol{y}_k \rangle \geq \alpha \cdot \epsilon \|\boldsymbol{s}_k\|$ with $\alpha > 2$ ensures positive-definiteness in the Hessian approximation.

### 3.2 Block-wise Approximation: Improve Memory Efficiency in Distributed Training

With the L-BFGS update rule, we need to store a sufficient number of history vectors $\boldsymbol{s}_k, \boldsymbol{y}_k$ to obtain a good Hessian approximation. Given model parameters $\boldsymbol{\theta} \in \mathbb{R}^d$ and supposing $M$ history vectors are needed, it required $O(2Md)$ memory space. While it is much smaller than storing the full Hessian ($O(d^2)$) with $M \ll d$, it can still be a critical bottleneck when training large models.

Furthermore, when it comes to distributed training, simply adopting data parallelism (DP) unfortunately does not help relieve memory pressure on each node, as parameters and gradients are duplicated on each node in DP. A naive implementation of the L-BFGS update needs to maintain a copy of $\{\boldsymbol{s}_k, \boldsymbol{y}_k\}_{k=1}^{M}$ on each node. As a result, more nodes do not help reduce per-node memory costs in the Hessian approximation. A more efficient design is needed to *reduce per-node memory costs when more nodes are available.*

To improve memory efficiency, we propose a new block-diagonal Hessian approximation. Unlike diagonal approximation methods such as KFAC (Ba et al., 2017), our block-wise approximation method allows each diagonal Hessian block to capture curvature information across multiple layers. Such a design enjoys a more flexible configuration depending on the size of each layer's parameters and each node's capacity. For instance, as shown in Figure 2, if node 1 and 2 have similar memory capacities, and the sizes of layers 1-2 and 3-4 are also close, we can approximate the Hessian blocks with layers 1-2 and 3-4 in node 1 and 2, respectively. Otherwise, we can group layers in a different way and ensure node 1 and 2 share similar memory costs.

During training, for the Hessian approximation, the $i$-th node only collects parameters and gradients corresponding to its block and computes and stores vectors $\{\boldsymbol{s}_k^i, \boldsymbol{y}_k^i\}$ based on Eq (6). When applying gradient conditioning, each node performs the Hessian-vector product on the corresponding layers, followed by updating parameters. And then, all nodes invoke a *All-Gather* operation to send/collect updated parameters to/from other nodes

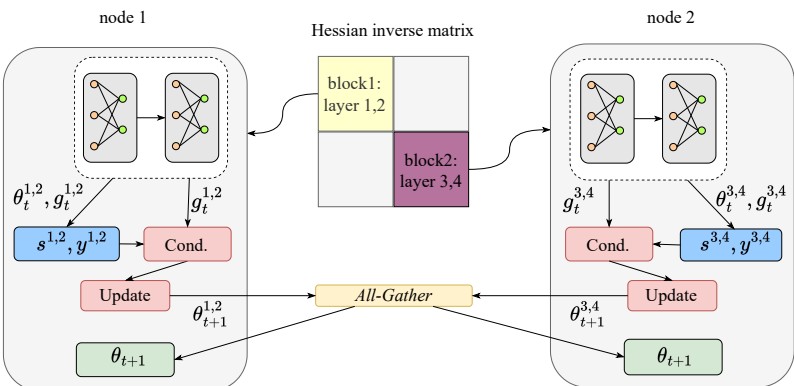

Figure 2: A distributed block-wise approximation. Each node can compute the Hessian inverse of a diagonal block comprising one or more layers. Unlike diagonal approximation methods such as KFAC, mL-BFGS allows much more flexible configurations depending on the size of each layer's parameters and each node's capacity.

**Analysis -** Given $p$ nodes and assuming a model's parameters can be evenly distributed among these nodes, then each node only needs $O(\frac{2M}{p}d)$ memory to store the history vectors. If more nodes are available, per-

node memory costs can be further reduced. On the other hand, it is easy to show that the Hessian inverse comprising block-diagonal matrices is positive-definite if each of these blocks is positive-definite, therefore ensuring stable convergence with positive-definite diagonal Hessian blocks.

**Lemma 3.** *In $k$-th Hessian update, if block $i(i = 1, \cdots, p)$, $\hat{H}_k^i$ is bounded by $\xi^i I \preceq \hat{H}_k^i \preceq \Xi^i I$, then $\min \xi^i \preceq \hat{H}_k \preceq \max \Xi^i$.*

### 3.3 Put All Together

With the momentum-based design and block-wise approximation, we present the whole algorithm, mL-BFGS, as shown in Alg 1. For each parameter block $i$ at iteration $t$, we compute momentum of parameters and gradients, and store them in $\mathcal{M}_{\boldsymbol{\theta}_t}$ and $\mathcal{M}_{\boldsymbol{g}_t}$ (line 6). For every $T$ iterations, we compute $\boldsymbol{s}_k^i, \boldsymbol{y}_i$, and update the Hessian approximation (line 13-18). It is worth noting that in *UpdateHessian*, the Hessian inverse is not explicitly computed. Instead, we only push $\boldsymbol{s}_k^i, \boldsymbol{y}_k^i$ to the history vector buffer. If the buffers are full, we will pop the oldest vectors. In the first $2T$ iterations, we use SGD to conduct a warmup training as the Hessian inverse is not available yet (line 8). After the initial warmup training, we will pre-condition gradients $\boldsymbol{g}_t$ before applying updates to $\boldsymbol{\theta}$ (line 10-11). At the end of each iteration, a *All-Gather* is called to update parameters on all nodes (line 20).

---

**Algorithm 1** mL-BFGS algorithm ($T$, $M$, parameter block $\{\boldsymbol{\theta}^i\}_{i=1}^p$)

---

1: Initialize $\boldsymbol{\theta}_0$, $\mathcal{M}_{\boldsymbol{\theta}_0}^i = \boldsymbol{\theta}_0^i$, $\mathcal{M}_{\boldsymbol{g}_0}^i = \boldsymbol{g}_0^i$

2: **for** $t = 1, \cdots, \text{max\_iter}$ **do**
3:      Randomly choose mini-batch input $\mathbf{X}_t \in \mathcal{X}$
4:      Perform model forward and backward, compute gradients $\boldsymbol{g}_t$ given $\mathbf{X}_t$
5:      **for** each parameter block $i$ **do**
6:          $\mathcal{M}_{\boldsymbol{\theta}_t}^i = \beta \cdot \mathcal{M}_{\boldsymbol{\theta}_{t-1}}^i + (1 - \beta) \cdot \boldsymbol{\theta}_t^i, \quad \mathcal{M}_{\boldsymbol{g}_t}^i = \beta \cdot \mathcal{M}_{\boldsymbol{g}_{t-1}}^i + (1 - \beta) \cdot \boldsymbol{g}_t^i$

7:          **if** $t \leq 2T$ **then**
8:              Warmup with SGD: $\boldsymbol{\theta}_{t+1}^i = \boldsymbol{\theta}_t^i - \eta_t \cdot \boldsymbol{g}_t^i$
9:          **else**
10:             Pre-condition: $\Delta\boldsymbol{\theta}_t^i = \hat{H}_k^i \cdot \boldsymbol{g}_t^i$ {Algorithm 2}
11:             $\boldsymbol{\theta}_{t+1}^i = \boldsymbol{\theta}_t^i - \eta_t \cdot \Delta\boldsymbol{\theta}_t^i$
12:          **end if**

13:          **if** $t\%T == 0$ and $t > T$ **then**
14:             $k = k + 1$
15:             $\boldsymbol{s}_k^i = \mathcal{M}_{\boldsymbol{\theta}_t}^i - \mathcal{M}_{\boldsymbol{\theta}_{t-T}}^i, \quad \boldsymbol{y}_k^i = \mathcal{M}_{\boldsymbol{g}_t}^i - \mathcal{M}_{\boldsymbol{g}_{t-T}}^i$
16:             Apply damping: $\hat{\boldsymbol{y}}_k^i = \tau \cdot \boldsymbol{y}_k^i + (1 - \tau) \cdot \boldsymbol{s}_k^i$
17:             Update Hessian: $\hat{H}_k^i = UpdateHessian(\hat{H}_{k-1}^i, \boldsymbol{s}_k^i, \hat{\boldsymbol{y}}_k^i, M)$
18:          **end if**

19:      **end for**

20:      *All-Gather* $\{\boldsymbol{\theta}_{t+1}^i\}_{i=1}^p$ across all nodes.
21: **end for**

---

**Hessian damping -** For non-convex QN optimization, damping is a common technique that ensures the positive definiteness of the Hessian (Al-Baali et al., 2014; Al-Baali & Grandinetti, 2017; Martens & Grosse, 2015). Furthermore, even with momentum, stochastic training can still cause undesirable fluctuation in the Hessian approximation. To preserve the positive of the Hessian, we adopt an adaptive damping scheme as

$$\hat{\boldsymbol{y}}_i = \tau \cdot \boldsymbol{y}_i + (1 - \tau) \cdot \boldsymbol{s}_i, \tag{7}$$

with $\tau$ obtained as

$$\tau = \begin{cases} \min(\frac{1-\sigma_L}{1-\mu}, \tau_0) & \mu \leq \sigma_L < 1 \\ \min(\frac{\sigma_H-1}{\mu-1}, \tau_0) & \mu \geq \sigma_H > 1 \\ \tau_0 & \text{otherwise} \end{cases}$$

where $\mu = \frac{\boldsymbol{s}_i^T \cdot \boldsymbol{y}_i}{\boldsymbol{s}_i^T \cdot \boldsymbol{s}_i}$, $\sigma_L$ and $\sigma_H$ are the lower and upper thresholds for restraining eigenvalues in $\hat{H}$ and $0 < \tau_0 < 1$ is a constant coefficient.

It is easy to show that $\frac{\boldsymbol{s}_i^T \cdot \hat{\boldsymbol{y}}_i}{\boldsymbol{s}_i^T \cdot \boldsymbol{s}_i}$ is bounded between $[\sigma_L, \sigma_H]$, so that the positive definiteness of the Hessian approximation is preserved during training (See A.2.4 for the proof).

## 4 Theoretical Guarantees

In this section, we first prove that mL-BFGS achieves a linear convergence rate under non-convex settings with proper assumptions. Then, in the second part of this section, we delve into the compute and memory costs in mL-BFGS, and show its benefits in wall-clock convergence compared to other baseline methods: stochastic L-BFGS, and KFAC.

### 4.1 Convergence Analysis

We assume the risk function $\mathcal{L}$ satisfies the following conditions:

*AS 1.* $\mathcal{L}(\boldsymbol{\theta})$ is twice continuously differentiable.

*AS 2.* $\ell_i(\boldsymbol{\theta})$ is $\Lambda$-smooth for $1 \leq i \leq N$, $\Lambda > 0$: $\forall \boldsymbol{\theta}_1, \boldsymbol{\theta}_2, \|\nabla \ell_i(\boldsymbol{\theta}_2) - \nabla \ell_i(\boldsymbol{\theta}_1)\| \leq \Lambda \|\boldsymbol{\theta}_2 - \boldsymbol{\theta}_1\|$.

*AS 3.* $\mathcal{L}(\boldsymbol{\theta})$ is $\lambda$-PL: it satisfies Polyak-Lojasiewicz (PL) condition for a constant $\lambda > 0$: $\|\nabla \mathcal{L}(\boldsymbol{\theta})\|^2 \geq \lambda \mathcal{L}(\boldsymbol{\theta})$.

The smooth condition in AS 2 is commonly used in analyzing convergence in practical optimization. In addition, noting that compared to typical strong convexity assumptions, the PL condition in AS 3 applies to a more general setting (Polyak, 1963). Strong convexity implies the PL condition, but not vice versa. In AS 3, we relax the constraint and only require the gradient variance to be lower bounded.

With the assumptions, we present the convergence theorem as follows. Proofs are deferred to Appendix A.2.

**Theorem 2.** *Assume AS 1-3 hold at each iteration t of mL-BFGS with mini-batch input $\mathbf{X}_t$ where each sample is randomly sampled from $\mathcal{X}$ with replacement, then the expectation of $\mathcal{L}(\boldsymbol{\theta}_t)$ satisfies*

$$E_{\mathbf{X}_t}[\mathcal{L}(\boldsymbol{\theta}_t)] \leq \alpha_{t-1} E_{\mathbf{X}_{t-1}}[\mathcal{L}(\boldsymbol{\theta}_{t-1})],$$

*where $\alpha_{t-1} = 1 - \eta_{t-1}\lambda\xi + \eta_{t-1}^2 \Lambda^2 \Xi^2$. $\xi$ and $\Xi$ denotes the lower and upper bound of the $\hat{H}$.*

By choosing $\eta_{t-1}$ such that $\alpha_{t-1} < 1$, mL-BFGS converges at a linear rate. The convergence rate matches the best rate in stochastic QN optimizations. It is worth mentioning that no convergence rate beyond linear is observed in stochastic L-BFGS optimizations. Theorem 2 is not aimed to push the theoretical convergence limit. Instead, it investigates the effects of momentum in the Hessian and block-diagonal approximation. In addition, as mentioned earlier, Theorem 2 also applies to convex settings, as strong convexity implies $\|\nabla \mathcal{L}(\boldsymbol{\theta})\|^2$ is lower bounded by $\mathcal{L}(\boldsymbol{\theta})$ for an appropriate $\lambda > 0$ and hence AS 3 holds.

### 4.2 Complexity Analysis

In this section, we analyse the compute and memory cost of SGD, KFAC(Ba et al., 2017), stochastic L-BFGS (we call it sL-BFGS)(Chang et al., 2019) and mL-BFGS. As the main motivation, reducing the complexities of QN methods is crucial for their deployment in real large-scale neural network optimization.

Given a model with parameter $\boldsymbol{\theta} \in \mathbb{R}^d$, we use $C_{\text{fb}}$ and $M_{\text{fb}}$ to represent the compute and memory cost of a forward/backward (Fwd/Bwd) pass with a batch size of $b = 1$. Furthermore, $C_{\text{opt}}$ denotes the compute cost of model updates (Opt) which consists of gradient reduction, computing and apply the update $\Delta\boldsymbol{\theta}$.

Table 1 summarizes the total compute and memory cost of SGD, KFAC, sL-BFGS and mL-BFGS in a general distributed system with $p$ workers. Compared to SGD, during the forward and backward passes, mL-BFGS needs to additionally compute $\mathcal{M}_{\boldsymbol{\theta}}$, $\mathcal{M}_{\boldsymbol{g}}$, for which the complexity increases linearly with model size ($d$). The main extra compute mL-BFGS introduces is the Hessian-vector product, in which we need to iterate over $\{\boldsymbol{s}_i\}_{i=1}^M$ and $\{\boldsymbol{y}_i\}_{i=1}^M$, as shown in Alg A.1. The complexity increases linearly with the number of history vectors and model size ($2Md$). However, it only adds average cost of $O(\frac{2Md}{p})$ on each worker. Compared to $O(bC_{\text{fb}})$ complexity in forward and backward passes, such costs are relatively marginal.

As a comparison, KFAC adds significant additional computations through 1) possible multiple backward passes to update factors ($\gamma bC_{\text{fb}}$ with $\gamma \geq 1$), 2) matrix inversion ($\sum(l_i^3 + (\frac{d_i}{l_i})^3)$) for every $T$ iterations, and 3) Matrix-vector products ($2\sum(l_i + \frac{d_i}{l_i})d_i$). On the other hand, sL-BFGS also resorts to computation-intensive operations including full-batch gradients and a separate large batch to estimate the Hessian, which respectively adds amortized costs of $O(bC_{\text{fb}})$ and $\frac{1}{T}b_H C_{\text{fb}}$. With data parallelism, it requires each worker to locally perform gradient conditioning, which adds another cost of $O(2Md)$ in total.

As for memory usage, mL-BFGS mainly needs $O(2Md)$ to store history vectors. The amortized costs of each worker are $O(\frac{2Md}{p})$. In practice, $M$ is set to be $10 \sim 20$, which ensures that memory usage is manageable in mL-BFGS. sL-BFGS needs $O(Md)$ storage for $\boldsymbol{s}_i, \boldsymbol{y}_i$ in total, and amortized cost of $O(\frac{1}{T}b_H M_{\text{fb}})$ for additional backward passes. While KFAC needs $O(2\sum(l_i^2 + (\frac{d_i}{l_i})^2))$ to store sub-matrices and their inverse, where the actual memory footprint hinges on model architectures.

Table 1: Computations and Memory in SGD, KFAC, sL-BFGS and mL-BFGS

|  | SGD | KFAC | sL-BFGS | mL-BFGS |
|---|---|---|---|---|
| | | Per-Node Computation | | |
| Fwd&Bwd | $O(bC_{\text{fb}})$ | $O(d + \gamma bC_{\text{fb}} + \frac{1}{T}\sum(l_i^3 + (\frac{d_i}{l_i})^3))$ | $O(d + 2bC_{\text{fb}} + \frac{1}{T}b_H C_{\text{fb}})$ | $O(\frac{d}{p} + bC_{\text{fb}})$ |
| Opt | $O(d)$ | $O(d + 2\sum(l_i + \frac{d_i}{l_i})d_i)$ | $O(d + 2Md)$ | $O(d + \frac{2Md}{p})$ |
| | | Per-Node Memory | | |
| Fwd&Bwd | $O(bM_{\text{fb}})$ | $O(d + bM_{\text{fb}})$ | $O(d + bM_{\text{fb}} + \frac{1}{T}b_H M_{\text{fb}})$ | $O(\frac{d}{p} + bM_{\text{fb}})$ |
| Opt | $O(d)$ | $O(d + 2\sum(l_i^2 + (\frac{d_i}{l_i})^2))$ | $O(d + 2Md)$ | $O(d + \frac{2Md}{p})$ |

- $b$: per-node batch size. $b_H$: batch size for the Hessian approx. $p$: #workers. $T$: Hessian update period.
- $d_i$: #params in $i$-th layer. $l_i$: #input neurons in $i$-th layer. $M$: max #history vectors.

Table 2 lists detailed amortized costs of different optimizers on ResNet-50/ImageNet in a distributed system. Compared to KFAC and sL-BFGS, mL-BFGS significantly reduces compute costs in approximating the Hessian and gradient conditioning. Due to the efficient distributed design, memory consumption on each worker is also reduced compared to sL-BFGS.

Table 2: Computation (MACs) and memory costs of different optimizers on ResNet-50/ImageNet ($b = 64$, $T = 20$; $M = 10$; $b_H = 1024$, $\gamma = 1$, $p = 8$). "B" denotes billion, and "M" denotes million.

|  | SGD | KFAC | sL-BFGS | mL-BFGS |
|---|---|---|---|---|
| Fwd/Bwd | | | 769B | |
| $\hat{H}$ Compute | - | 570B | 1414B | 52M |
| Opt | 26M | 156B | 4B | 546M |
| $\hat{H}$ Memory | - | 308M | 4B | 520M |

## 5 Experiments

We conduct various experiments on computer vision (CV) problems involving datasets such as CIFAR-10, CIFAR-100 and ImageNet. We choose SGD and Adam as the main baselines since it is widely used in these tasks and maintains the best training performance. We also compare with another quasi-Newton method,

KFAC, in large-scale model training. We separately tune hyperparameters for each optimizer to ensure it achieves the best validation accuracy.

We use a single GPU server with 8 Nvidia Quadro RTX 5000 GPUs to simulate a distributed system, where each GPU is used as a worker to perform forward and backward passes, and model updates. Furthermore, each worker is also assigned with one Hessian block to compute the Hessian inverse and gradient conditioning. The current implementation is based on PyTorch. We set lower and upper thresholds of damping $\sigma_L, \sigma_H$ to be $0.01, 1.5$ in all experiments to smooth the Hessian approximation.

### 5.1 Experiments on CIFAR-10/CIFAR-100

We first evaluate mL-BFGS on two small-scale problems: CIFAR-10 and CIFAR-100, and demonstrate the convergence advantage of mL-BFGS compared to SGD and Adam. The models used are ResNet-18 and DeiT-Tiny Touvron et al. (2021), where DeiT-Tiny is an efficient image transformer with 12 layers, 3 attention heads, and hidden and MLP dimension of 192.

For ResNet18, we divide it into 4 blocks such that each block consists of 2 *resblocks* (He et al. (2016)). The linear layer for classification is packed into the last block. For DeiT-Tiny, due to the small model size, we choose to approximate the whole Hessian.

Hyperparameters are tuned to achieve the best validation accuracy. Details are provided in Appendix A.4.1.

Figure 3 shows training loss. We observe that mL-BFGS achieves a much faster convergence rate compared to SGD and ADAM. Table 3 lists the validation accuracy on CIFAR-10 and CIFAR-100. We note that mL-BFGS also achieves similar accuracy as SGD. On the other hand, as in Figure 3, although ADAM has a convergence rate close to mL-BFGS, the validation accuracy are much lower in all experiments compared to mL-BFGS. Therefore, we obverse mL-BFGS not only deliver faster convergence, but also achieves good generalization performance.

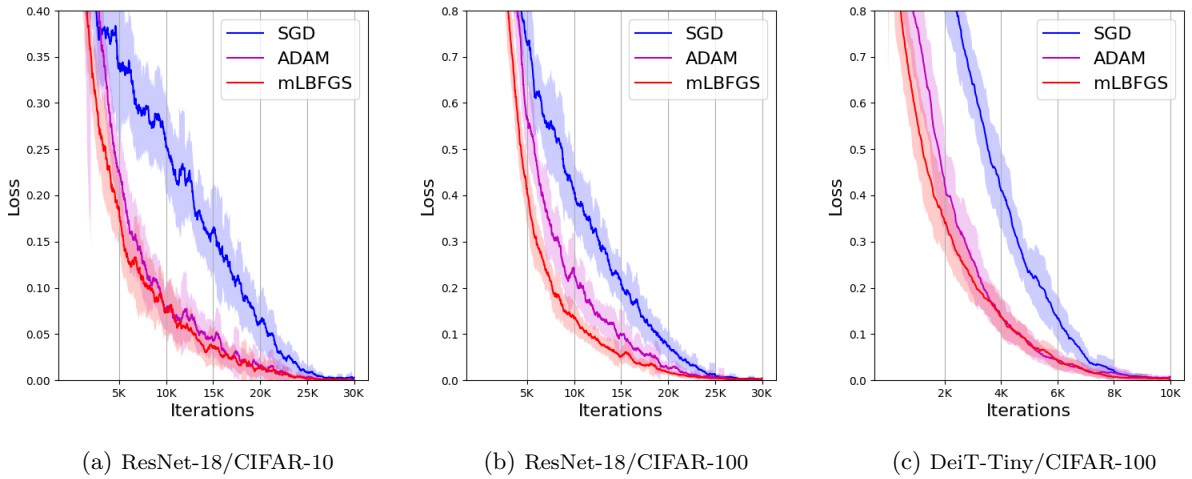

(a) ResNet-18/CIFAR-10  (b) ResNet-18/CIFAR-100  (c) DeiT-Tiny/CIFAR-100

Figure 3: Training loss mL-BFGS, SGD, ADAM on ResNet-18 and Deit-Tiny on CIFAR-10/100. mL-BFGS delivers faster convergence than SGD and ADAM.

Table 3: Validation accuracy of ResNet-18 and Deit-Tiny on CIFAR-10/100 using SGD, ADAM, and mL-BFGS.

| ResNet-18/CIFAR-10 | | | ResNet-18/CIFAR-100 | | | DeiT-Tiny/CIFAR-100 | | |
|---|---|---|---|---|---|---|---|---|
| SGD | ADAM | mL-BFGS | SGD | ADAM | mL-BFGS | SGD | ADAM | mL-BFGS |
| $94.1 \pm 0.1$ | $92.7 \pm 0.1$ | $93.9 \pm 0.1$ | $75 \pm 0.15$ | $72.2 \pm 0.16$ | $74.4 \pm 0.1$ | $80.5 \pm 0.2$ | $75.3 \pm 0.3$ | $79.9 \pm 0.2$ |

## 5.2 Experiments on ImageNet

ImageNet has been the gold standard for evaluating the performance of optimizers. It consists of $\sim$1.2M training and $\sim$50K test images, categorized into 1000 classes. We follow the standard data pre-processing procedure, where each image is first resized to $256 \times 256$, and randomly cropped to $224 \times 224$ and flipped horizontally. Each image is then normalized using pre-computed mean and variance.

**ResNet-50** – When approximating the Hessian, we divide ResNet-50 into 8 blocks such that each block consists of 2 *resblocks*. Similar to ResNet-18 in CIFAR-10, the linear layer is packed into the last block. Figure 4a shows iteration-wise convergence on ResNet-50 using SGD, Adam, KFAC and mL-BFGS. Detailed hyperparameter settings are provided in Appendix A.4.2. Compared to Adam and SGD, mL-BFGS enjoys much faster per-iteration convergence. Such fast convergence is also reflected in the validation dataset (Figure 4b). Furthermore, it also generalizes well on the validation set, and finally reaches comparable validation accuracy to SGD.

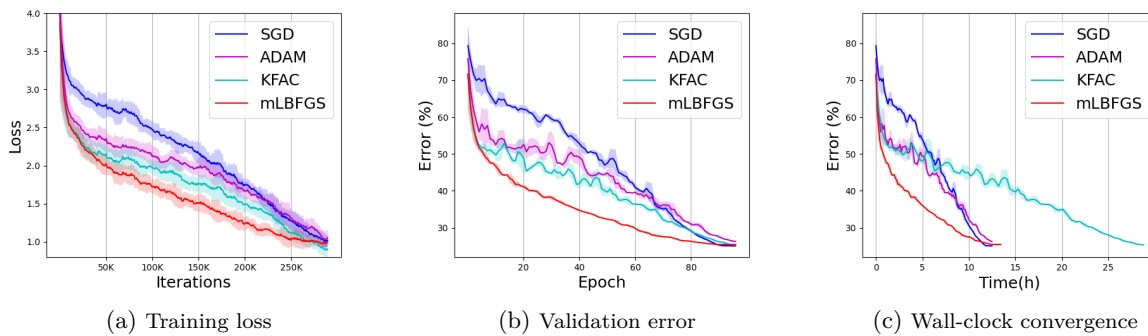

|  (a) Training loss  |  (b) Validation error  |  (c) Wall-clock convergence  |

Figure 4: Training loss and validation error of mL-BFGS, SGD, Adam and KFAC on ImageNet using ResNet-50. mL-BFGS delivers faster iteration-wise and real-time convergence compared to SGD and Adam. We plot the mean and standard error over 3 runs with different random seeds

Table 4: Validation accuracy of ResNet-50 on ImageNet using SGD, ADAM, KFAC, and mL-BFGS.

| SGD | | ADAM | | mL-BFGS | | KFAC | |
|---|---|---|---|---|---|---|---|
| Acc | Time/epoch | Acc | Time/epoch | Acc | Time/epoch | Acc | Time/epoch |
| $74.9 \pm 0.11$ | 7.8min | $73.95 \pm 0.1$ | 7.8min | $74.6 \pm 0.13$ | 7.9min | $74.5 \pm 0.1$ | 17min |

The benefit of mL-BFGS is even more striking in terms of wall-clock time. As listed in Figure 4c and Table 4, due to light compute costs, the per-epoch runtime of mL-BFGS is almost the same as SGD and Adam. On the other hand, for KFAC, while it delivers fast per-iteration convergence compared to SGD and ADAM, the wall-clock performance is significantly diminished by its additional compute costs. The per-epoch runtime is $> 2\times$ more than mL-BFGS.

## 5.3 Ablation Study: The Effects of Momentum and Damping

In this section, we give more insight into the effects of momentum and damping used in mL-BFGS. To this end, we ablate two critical components in mL-BFGS: momentum and damping in the Hessian approximation, and then use the ablated version to train ResNet-18 on CIFAR-10. We focus on CIFAR-10 since we observed more convergence instability on this dataset compared to others.

Figure 5 shows convergence using the ablated mL-BFGS with only momentum (black), with only damping (purple), and with no momentum or damping (red). Due to stochastic noise, the ablated version of mL-BFGS without momentum/damping (vanilla L-BFGS) diverges easily in the early stages. With momentum (black), the whole optimization is significantly stabilized. However, it still fails to converge when there is a radical change in the loss landscape (for example, when learning rate decays). With damping (purple), the Hessian

approximation is effectively restrained, especially when such sudden changes in the loss landscape happen. It is interesting to observe that while damping prevents divergence, the whole training is still largely affected by stochastic noise. Notable fluctuation in the loss is commonly observed during training. As a comparison, the complete mL-BFGS (blue) effectively addresses these issues achieving much more stable convergence.

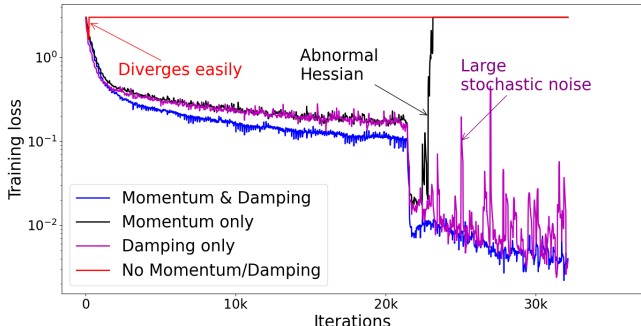

Figure 5: Ablation study for mL-BFGS on ResNet-18/CIFAR-10 (batch size: 256). Vanilla L-BFGS (red) diverges easily. Damping-only scheme (purple) cannot effectively suppress stochastic noise. Momentum-only scheme (black) still diverges with radical changes in the Hessian. While damping and momentum combined (blue) achieve a very smooth and stable optimization.

## 6    Related Works

While SGD is widely used in many machine learning tasks, other forms of optimization have also been investigated extensively in the past years. Among these attempts, designing optimizers with preconditioned gradients is one of the most promising areas.

Adaptive methods such as Adam, AdaGrad, AdaDelta (Kingma & Ba, 2014; Duchi et al., 2011; Zeiler, 2012) construct a diagonal matrix by incorporating knowledge from the past gradients. Such a diagonal matrix adaptively adjusts the learning rate for each parameter. For instance, AdaGrad uses a large learning rate for those irrelevant features (small gradients), and small learning for those relevant ones (large gradients). Adams further uses the first and second moment of gradients to adjust the learning rate.

Besides diagonal preconditioning matrices, constructing block diagonal matrices or even full matrices has received increasing attention in recent years. Methods such as Shampoo (Gupta et al., 2018) approximate the full matrix version of AdaGrad as a block-diagonal matrix to incorporate more curvature information during optimization. Similarly, a well-known KFAC method (Martens & Grosse, 2015) approximates the Fisher information matrix as a block-diagonal matrix. L-BFGS methods Moritz et al. (2016) on the other hand directly construct the full Hessian matrix as a preconditioner during optimization. These preconditioners have been empirically proved to achieve fast convergence compared to SGD, as well as adaptive methods. Authors in Goldfarb et al. (2020) further adopt L-BFGS to efficiently compute matrix inversion in KFAC.

Variance reduction in the preconditioning matrix is also crucial to ensure stable optimization. The algorithm in Moritz et al. (2016) adopts a separate large batch of data to estimate current curvature. On the other hand, VITE Lucchi et al. (2015) chooses to use a pivot parameter together with full-batch gradients to reduce variance in the Hessian approximation.

## 7    Conclusion

In this paper, we propose mL-BFGS, a quasi-Newton method that simultaneously mitigates computation and convergence instability barriers in second-order methods, as well as scalability issues in distributed systems. By introducing momentum and damping into the Hessian update, mL-BFGS obviates the need for highly costly estimation on the Hessian. Approximation along diagonal blocks further reduces memory and compute costs in distributed systems. Empirical analyses on CV models, such as ResNet-50 and Vision Transformer show that mL-BFGS achieves faster convergence, and reaches similar accuracy compared to SGD.

## Acknowledgements

This material is based upon work supported by Defense Advanced Research Projects Agency (DARPA) under Contract FASTNICS HR001120C0088. The views, opinions, and/or findings expressed are those of the author(s) and should not be interpreted as representing the official views or policies of the Department of Defense or the U.S. Government.

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

# A   Appendix

The appendix is arranged as follows: In Sec A.1, we include the Hessian-vector product used in mL-BFGS. In Sec A.2 and A.3, we provide proofs of lemmas and theorems in the paper. In Sec A.4, we list hyperparameters used in the experiments.

## A.1   Hessian-Vector Product in L-BFGS

---
**Algorithm 2** Hessian-Vector in L-BFGS

---
**Input:** $\boldsymbol{g}_t$, $\{\boldsymbol{y}_i\}_{i=1}^{M}$, $\{\boldsymbol{s}_i\}_{i=1}^{M}$
**Output:** $\boldsymbol{g}_t$
 1: **for** $i = 0, \cdots, M-1$ **do**
 2:     $\rho_i = \boldsymbol{s}_i^T \cdot \boldsymbol{y}_i$
 3: **end for**
 4: **for** $i = 0, \cdots, M-1$ **do**
 5:     $\alpha_i = \frac{\boldsymbol{s}_{M-i-1}^T \boldsymbol{g}_t}{\rho_{M-i-1}}$
 6:     $\boldsymbol{g}_t = \boldsymbol{g}_t - \alpha_i \cdot \boldsymbol{y}_{M-i-1}$
 7: **end for**
 8: $\boldsymbol{g}_t = \hat{H}_0 \cdot \boldsymbol{g}_t$ $\{\hat{H}_0 = \frac{\boldsymbol{s}_{M-1}^T \boldsymbol{y}_{M-1}}{\boldsymbol{y}_{M-1}^T \boldsymbol{y}_{M-1}} \cdot I\}$
 9: **for** $i = 0, \cdots, M-1$ **do**
10:     $\beta_i = \frac{\boldsymbol{y}_i^T \boldsymbol{g}_t}{\rho_i}$
11:     $\boldsymbol{g}_t = \boldsymbol{g}_t + (\alpha_{M-i-1} - \beta_i) \cdot \boldsymbol{s}_i$
12: **end for**

---

## A.2   Proof of Lemmas and Theorems

### A.2.1   Proof of Lemma 1

*Proof.* Let $\mathcal{M}_{\boldsymbol{n}_t}$ denotes the stochastic noise in $\mathcal{M}_{\boldsymbol{g}_t}$, then it can be written as:

$$\mathcal{M}_{\boldsymbol{n}_t} = \beta \mathcal{M}_{\boldsymbol{n}_{t-1}} + (1-\beta)\boldsymbol{n}_t = \beta^t \boldsymbol{n}_0 + (1-\beta)\sum_{i=1}^{t} \beta^{(t-i)}\boldsymbol{n}_i$$

Since $\boldsymbol{n}_i$ for $i = 0, \cdots, t$ are independent, therefore

$$E\left[\|\mathcal{M}_{\boldsymbol{n}_t}\|^2\right] = \beta^{2t} E\left[\|\boldsymbol{n}_0\|^2\right] + \sum_{i=1}^{t}(1-\beta)^2 \beta^{2(t-i)} E\left[\|\boldsymbol{n}_i\|^2\right]$$

Since $E\left[\|\boldsymbol{n}_i\|^2\right]$ is bounded by $\epsilon^2$, therefore

$$E\left[\|\mathcal{M}_{\boldsymbol{n}_t}\|^2\right] \le \beta^{2t}\epsilon^2 + \sum_{i=1}^{t}(1-\beta)^2 \beta^{2(t-i)}\epsilon^2 = \epsilon^2(\beta^{2t} + \tfrac{(1-\beta)^2}{1-\beta^2}(1-\beta^{2t}))$$

It is obvious that $\lim_{t\to\infty} E\left[\|\mathcal{M}_{\boldsymbol{n}_t}\|^2\right] \le \frac{1-\beta}{1+\beta}\epsilon^2$.

Furthermore, noise variance in $\boldsymbol{y}_k$ is bounded by $\lim_{t\to\infty} E\left[\|\mathcal{M}_{\boldsymbol{n}_t} - \mathcal{M}_{\boldsymbol{n}_{t-L}}\|^2\right] \le \frac{4(1-\beta)}{1+\beta}\epsilon^2$. $\qquad\square$

### A.2.2 Proof of Lemma 2

*Proof.* First, consider the case with $T = 1$, then

$$\begin{aligned}
\boldsymbol{s}_0 &= \mathcal{M}_{\boldsymbol{\theta}_1} - \mathcal{M}_{\boldsymbol{\theta}_0} = (1-\beta)(\boldsymbol{\theta}_1 - \boldsymbol{\theta}_0) \\
\boldsymbol{y}_0 &= \mathcal{M}_{\boldsymbol{g}_1} - \mathcal{M}_{\boldsymbol{g}_0} = (1-\beta)(\boldsymbol{g}_1 - \boldsymbol{g}_0)
\end{aligned}$$

Since $\boldsymbol{g}_1 - \boldsymbol{g}_0 = B(\boldsymbol{\theta}_1 - \boldsymbol{\theta}_0)$ in L-BFGS update, then it is also valid that $\boldsymbol{y}_0 = B\boldsymbol{s}_0$.

Assume for $(k-1)$th approximation, $\boldsymbol{y}_k = \mathcal{M}_{\boldsymbol{g}_k} - \mathcal{M}_{\boldsymbol{g}_{k-1}} = B\boldsymbol{s}_k = B(\mathcal{M}_{\boldsymbol{\theta}_k} - \mathcal{M}_{\boldsymbol{\theta}_{k-1}})$.

Expand $\mathcal{M}_{\boldsymbol{\theta}_k}$, we can easily get $\boldsymbol{y}_k = \mathcal{M}_{\boldsymbol{g}_k} - \mathcal{M}_{\boldsymbol{g}_{k-1}} = (1-\beta)B(\boldsymbol{\theta}_k - \mathcal{M}_{\boldsymbol{\theta}_{k-1}})$.

Then, for $k$th approximation, we have

$$B\boldsymbol{s}_{k+1} = B(\mathcal{M}_{\boldsymbol{\theta}_{k+1}} - \mathcal{M}_{\boldsymbol{\theta}_k}) = B(1-\beta)(\boldsymbol{\theta}_{k+1} - \mathcal{M}_{\boldsymbol{\theta}_k})$$

Further expand $\mathcal{M}_{\boldsymbol{\theta}_k}$, we get

$$B\boldsymbol{s}_{k+1} = (1-\beta)B(\boldsymbol{\theta}_{k+1} - \boldsymbol{\theta}_k + \beta(\boldsymbol{\theta}_k - \mathcal{M}_{\boldsymbol{\theta}_{k-1}}))$$

Similarly, since $\boldsymbol{g}_{k+1} - \boldsymbol{g}_k = B(\boldsymbol{\theta}_{k+1} - \boldsymbol{\theta}_k)$, we have

$$B\boldsymbol{s}_{k+1} = (1-\beta)(\boldsymbol{g}_{k+1} - \boldsymbol{g}_k) + \beta(\mathcal{M}_{\boldsymbol{g}_k} - \mathcal{M}_{\boldsymbol{g}_{k-1}}) = \boldsymbol{y}_k.$$

Consider the case with $T > 1$, $\boldsymbol{y}_k$ can be written as

$$\boldsymbol{y}_k = \sum_{i=0}^{T-1} \mathcal{M}_{\boldsymbol{g}_{g(k+1)T-i}} - \mathcal{M}_{\boldsymbol{g}_{g(k+1)T-i-1}}$$

According to the result with $T = 1$, $\boldsymbol{y}_k$ can be further written as

$$\boldsymbol{y}_k = \sum_{i=0}^{T-1} B(\mathcal{M}_{\boldsymbol{\theta}_{g(k+1)T-i}} - \mathcal{M}_{\boldsymbol{\theta}_{g(k+1)T-i-1}}) = B\sum_{i=0}^{T-1}(\mathcal{M}_{\boldsymbol{\theta}_{g(k+1)T-i}} - \mathcal{M}_{\boldsymbol{\theta}_{g(k+1)T-i-1}}) = B\boldsymbol{s}_k.$$

Therefore, it is equivalent to use the momentum based scheme to estimate the Hessian without losing accuracy compared to naive L-BFGS.

$\qquad\square$

### A.2.3  Proof of Theorem 1

*Proof.* First, we show that $\frac{\langle s_k, y_k \rangle}{\langle s_k, s_k \rangle}$ is well bounded, so that most iterates using BFGS goes toward desirable direction according to Byrd & Nocedal (1989).

Given $\langle s_k, y_k \rangle \geq \alpha \cdot \epsilon \|s_k\|$, we have $\frac{\langle s_k, y_k \rangle}{\langle s_k, s_k \rangle} \geq \frac{\alpha \epsilon}{\|s_k\|}$.

Let $v_k$ denotes gradient changes with no noise added, then $y_k = v_k + n_k - n_{k-1}$. Since noise variance in $y_k$ is bounded by $2\epsilon^2$, we have $E\left[\|n_k\|^2\right] \leq \epsilon^2$.

Then,

$$E\left[\frac{\langle s_k, y_k \rangle}{\langle s_k, s_k \rangle}\right] = E\left[\frac{\langle s_k, v_k \rangle}{\langle s_k, s_k \rangle} - \frac{\langle s_k, n_k - n_{k-1} \rangle}{\langle s_k, s_k \rangle}\right] \geq \frac{\langle s_k, v_k \rangle}{\langle s_k, s_k \rangle} - \frac{2\epsilon}{\|s_k\|} \geq \lambda - \frac{2\epsilon}{\|s_k\|}$$

According to the two inequalities above, we have $E\left[\frac{\langle s_k, y_k \rangle}{\langle s_k, s_k \rangle}\right] \geq \frac{\alpha}{\alpha+2}\lambda$.

As for the upper bound, first have $E\left[\langle s_k, y_k \rangle\right] = E\left[\langle s_k, vk + n_k - n_{k-1} \rangle\right] \leq \langle s_k, vk \rangle + 2\epsilon\|s_k\|$.

Because $H \preceq \Lambda I$, then $\langle s_k, vk \rangle \leq \Lambda\|s_k\|^2$. Combined with $\langle s_k, y_k \rangle \geq \alpha\epsilon\|s_k\|$, we have

$$\|s_k\| \geq \frac{\alpha - 2}{\Lambda}\epsilon, \quad \langle s_k, y_k \rangle \leq \|s_k\|\left(\Lambda\|s_k\| + 2\epsilon\right) \tag{8}$$

.

With $\lambda I \preceq H \preceq \Lambda I$, we have

$$\langle s_k, vk \rangle \geq \frac{\lambda\Lambda}{\lambda+\Lambda}\|s_k\|^2 + \frac{1}{\lambda+\Lambda}\|vk\|^2$$

Rearrange this inequality, we have

$$\left\|vk - \frac{\lambda+\Lambda}{2}s_k\right\| \leq \frac{\Lambda-\lambda}{2}\|s_k\|$$

Due to $y_k = vk + n_k - n_{k-1}$, we can convert the equation above to

$$E\left[\left\|y_k - \frac{\lambda+\Lambda}{2}s_k\right\|^2\right] \leq \left(\frac{\Lambda-\lambda}{2}\|s_k\| + 2\epsilon\right)^2.$$

Expand the equation, we have

$$E\left[\|y_k\|^2 - (\Lambda + \lambda)\langle s_k, y_k \rangle + \left(\frac{\Lambda+\lambda}{2}\right)^2\|s_k\|^2\right] \leq \left(\frac{\Lambda-\lambda}{2}\right)^2\|s_k\|^2 + 2(\Lambda - \lambda)\|s_k\|\epsilon + 4\epsilon^2$$

Divide by $\langle s_k, s_k \rangle$ on both sides, we have

$$E\left[\frac{\langle y_k, y_k \rangle}{\langle s_k, s_k \rangle}\right] \leq E\left[\frac{(\Lambda+\lambda)\langle s_k, y_k \rangle}{\langle s_k, s_k \rangle}\right] + \frac{(2\epsilon + \Lambda\|s_k\|)(2\epsilon - \lambda\|s_k\|)}{\langle s_k, s_k \rangle}$$

According to Eq (8), we simplify the expectation as

$$E\left[\frac{\langle y_k, y_k \rangle}{\langle s_k, s_k \rangle}\right] \leq \frac{(\Lambda + \lambda)(\Lambda\|s_k\| + 2\epsilon)}{\|s_k\|} + \frac{(2\epsilon + \Lambda\|s_k\|)(2\epsilon - \lambda\|s_k\|)}{\langle s_k, s_k \rangle}$$

$$= \Lambda(\Lambda + \lambda) + \frac{2(\Lambda + \lambda)\epsilon}{\|s_k\|} + \frac{4\epsilon^2}{\|s_k\|^2} + \frac{2\Lambda\epsilon}{\|s_k\|} - \frac{2\lambda\epsilon}{\|s_k\|} - \Lambda\lambda$$

$$= \Lambda^2 + \frac{4\Lambda\epsilon}{\|s_k\|} + \frac{4\epsilon^2}{\|s_k\|^2}$$

$$\leq \Lambda^2 + \frac{4\Lambda^2}{\alpha - 2} + \frac{4\Lambda^2}{(\alpha - 2)^2} = \left(\frac{\alpha}{\alpha - 2}\Lambda\right)^2$$

With the well-bounded Hessian approximation, we show that there is an lower bound for the optimization with noise involved.

With $\langle \boldsymbol{s}_k, \boldsymbol{y}_k \rangle \geq \alpha \epsilon \|\boldsymbol{s}_k\|$ and $E\left[\|\boldsymbol{n}_t\|^2\right] \leq \epsilon^2$, we get $\langle \boldsymbol{s}_k, \boldsymbol{y}_k \rangle = \langle \boldsymbol{s}_k, \boldsymbol{v}k + \boldsymbol{n}_{t+1} - \boldsymbol{n}_t \rangle \leq \langle \boldsymbol{s}_k, \boldsymbol{v}k \rangle + 2\epsilon \|\boldsymbol{s}_k\|$.

Then $\langle \boldsymbol{s}_k, \boldsymbol{v}k \rangle \geq (\alpha - 2)\epsilon \|\boldsymbol{s}_k\|$.

With the upper bound of $H$, we have $(\alpha - 2)\epsilon \|\boldsymbol{s}_k\| \leq \Lambda \|\boldsymbol{s}_k\|^2$.

With the lower bound of $H$, we have $\mathcal{L}(\boldsymbol{\theta}_t) - \mathcal{L}(\boldsymbol{\theta}_{t-1}) \geq g_{t+1}(-\boldsymbol{s}_k) + \frac{\lambda}{2} \|\boldsymbol{s}_k\|^2$.

We focus on near-optimal region, where $\mathcal{L}(\boldsymbol{\theta}_{t+1}) = 0$ and $g_{t+1} = \boldsymbol{0}$, then we get $\mathcal{L}(\boldsymbol{\theta}_t) \geq \frac{\lambda}{2} \|\boldsymbol{s}_k\|^2 \geq \frac{(\alpha-2)^2 \lambda}{2\Lambda^2} \epsilon^2$.

Therefore, the achievable loss by L-BFGS with noise is lower bounded by $\frac{(\alpha-2)^2 \lambda}{2\Lambda^2} \epsilon^2$. $\qquad \square$

### A.2.4 Proof of Hessian Damping in Eq (7)

*Proof.* According to Eq (7), $\boldsymbol{s}_i^T \hat{\boldsymbol{y}}_i = \boldsymbol{s}_i^T (\tau \boldsymbol{y}_i + (1 - \tau) \boldsymbol{s}_i) = (\mu \tau + 1 - \tau) \boldsymbol{s}_i^T \boldsymbol{s}_i$, where $\mu = \frac{\boldsymbol{s}_i^T \boldsymbol{y}_i}{\boldsymbol{s}_i^T \boldsymbol{s}_i}$.

For $\mu \leq \sigma_L$, two cases need to be considered:

If $\tau = \tau_0$, then $\frac{1 - \sigma_L}{1 - \mu} \geq \tau_0$, and $\mu \tau + 1 - \tau \geq \sigma_L$.

If $\tau = \frac{1 - \sigma_L}{1 - \mu}$, then $\mu \tau + 1 - \tau = \sigma_L$

Therefore, when $\mu \leq \sigma_L$, $\boldsymbol{s}_i^T \hat{\boldsymbol{y}}_i \geq \sigma_L \boldsymbol{s}_i^T \boldsymbol{s}_i$

For $\sigma_L < \mu < \sigma_H$:

We can write $\mu \tau + 1 - \tau = \mu \tau_0 + 1 - \tau_0$. It is easy to show that

$$\mu \tau_0 + 1 - \tau_0 - \sigma_L \geq (1 - \sigma_L)(1 - \tau_0) > 0$$
$$\mu \tau_0 + 1 - \tau_0 - \sigma_H \leq (1 - \sigma_H)(1 - \tau_o) < 0$$

Therefore, when $\sigma_L < \mu < \sigma_H$, $\sigma_L \boldsymbol{s}_i^T \boldsymbol{s}_i < \boldsymbol{s}_i^T \hat{\boldsymbol{y}}_i < \sigma_H \boldsymbol{s}_i^T \boldsymbol{s}_i$.

For $\mu \geq \sigma_H$, similarly two cases might arise:

If $\tau = \tau_0$, then $\frac{\sigma_H - 1}{\mu - 1} \geq \tau_0$, and $\mu \tau + 1 - \tau \leq \sigma_H$.

If $\tau = \frac{\sigma_H - 1}{\mu - 1}$, then $\mu \tau + 1 - \tau = \sigma_H$.

Therefore, when $\mu \geq \sigma_H$, $\boldsymbol{s}_i^T \hat{\boldsymbol{y}}_i \leq \sigma_H \boldsymbol{s}_i^T \boldsymbol{s}_i$

In summary, $\sigma_L \leq \frac{\boldsymbol{s}_i^T \cdot \hat{\boldsymbol{y}}_i}{\boldsymbol{s}_i^T \cdot \boldsymbol{s}_i} \leq \sigma_H$. $\qquad \square$

### A.2.5 Proof of Lemma 3

*Proof.* For any arbitrary vector $\boldsymbol{x} \neq \boldsymbol{0}$, $\boldsymbol{x}^T \cdot \hat{H}_k \cdot \boldsymbol{x}$ can be written as

$(\boldsymbol{x}^{1^T} \cdot \hat{H}_k^1, \cdots, \boldsymbol{x}^{p^T} \cdot \hat{H}_k^p) \cdot (\boldsymbol{x}^{1^T}, \cdots, \boldsymbol{x}^{p^T})^T = \sum_{i=1}^p \boldsymbol{x}^{i^T} \cdot \hat{H}_k^i \cdot \boldsymbol{x}^i$,

where $\boldsymbol{x}^i$ is a sub-vector corresponding to block $i$.

Let $\xi \equiv \min \xi^i$ and $\Xi \equiv \max \Xi^i$, therefore, $\xi \leq \boldsymbol{x}^T \cdot \hat{H}_k \cdot \boldsymbol{x} \leq \Xi$. $\qquad \square$

### A.3 Proof of Theorem 2

To prove Theorem 2, we need several lemmas to bound the Hessian approximation and the gradient variance of the objective function.

#### A.3.1 Some Lemmas for Theorem 2

**Lemma 4.** *Given damping scheme in Eq (7), at the $k$-th Hessian update, $\hat{H}_k$ during the optimization is bounded by $\xi I \preceq \hat{H}_k \preceq \Xi I$, where $\Xi = (M+1)\frac{1}{\sigma_L}$, and $\xi = \frac{1}{\sigma_H}$.*

*Proof.* **Lower bound**: $\hat{H}$ is initialized as $\hat{H}_0 = \frac{s_0^T \hat{y}_0}{\hat{y}_0^T \hat{y}_0} \cdot I$. According to the damping scheme A.2.4, there exists $H_0(\boldsymbol{\theta}) \preceq \sigma_H I$ such that $\hat{y}_0 = H_0 \cdot s_0$.

Therefore, $\frac{s_0^T \hat{y}_0}{\hat{y}_0^T \hat{y}_0} \cdot I = \frac{s_0^T H_0 s_0}{s_0^T H_0 \cdot H_0 s_0} \cdot I = \frac{(s_0^T H_0^{1/2})(H_0^{1/2} s_0)}{(s_0^T H_0^{1/2}) \cdot H_0 \cdot (H_0^{1/2} s_0)} \cdot I \succeq \frac{1}{\sigma_H} I$.

Then for $k \geq 1$, assuming $\hat{H}_{k-1} \succeq \frac{1}{\sigma_H} I$ hold, based on Eq (4), $\hat{H}_k = (I - \rho_k \hat{y}_k s_k^T)^T \hat{H}_{k-1}(I - \rho_k \hat{y}_k s_k^T) + \frac{s_k s_k^T}{s_k^T \hat{y}_k}$.

Because $(I - \rho_k \hat{y}_k s_k^T)^T \hat{H}_{k-1}(I - \rho_k \hat{y}_k s_k^T)$ is positive definite, we can bound $\hat{H}_k$ as: $\hat{H}_k \succeq \frac{s_k s_k^T}{s_k^T \hat{y}_k} = \frac{s_k s_k^T}{s_k^T \cdot H_k \cdot s_k} \succeq \frac{1}{\sigma_H} I$

Therefore, lower bound of $\hat{H}_k$, $\xi = \frac{1}{\sigma_H}$.

**Upper bound**: Since $H_0(\boldsymbol{\theta}) \succeq \sigma_L$, we can get $\frac{s_0^T \hat{y}_0}{\hat{y}_0^T \hat{y}_0} \cdot I = \frac{(s_0^T H_0^{1/2})(H_0^{1/2} s_0)}{(s_0^T H_0^{1/2}) \cdot H_0 \cdot (H_0^{1/2} s_0)} \cdot I \preceq \frac{1}{\sigma_L}$.

Similarly, for $k \geq 1$, we assume $\hat{H}_{k-1} \preceq k \frac{1}{\sigma_L}$ hold.

For the first part in Eq (4), let $Q = (I - \rho_k \hat{y}_k s_k^T)$. Then for $\forall\, \boldsymbol{x} \neq \boldsymbol{0}$, $\boldsymbol{x}^T \cdot Q^T \hat{H}_{k-1} Q \cdot \boldsymbol{x} = (Q\boldsymbol{x})^T \cdot \hat{H}_{k-1} \cdot (Q\boldsymbol{x}) \leq \frac{k}{\sigma_L} \boldsymbol{x}^T \cdot (Q^T Q) \cdot \boldsymbol{x}$.

Let $P = \frac{s_k \hat{y}_k^T \hat{y}_k s_k^T}{s_k^T \hat{y}_k s_k^T \hat{y}_k} - \frac{\hat{y}_k s_k^T + s_k \hat{y}_k^T}{s_k^T \hat{y}_k}$, then $Q^T Q = I + P$.

Because $P$ is rank-1 matrix with eigenvalue $-1$ and eigenvector $\hat{y}_k$, we have $\boldsymbol{x}^T \cdot Q^T \hat{H}_{k-1} Q \cdot \boldsymbol{x} \leq \frac{k}{\sigma_L} \boldsymbol{x}^T \cdot (I + P) \cdot \boldsymbol{x} \leq \frac{k}{\sigma_L} \boldsymbol{x}^T \boldsymbol{x}$

For the second part in Eq (4), we can directly get $\frac{s_k s_k^T}{s^T y_k} = \frac{s_k s_k^T}{s_k^T \cdot H_k \cdot s_k} \preceq \frac{1}{\sigma_L} I$.

Therefore, $\boldsymbol{x}^T \cdot \hat{H}_k \cdot \boldsymbol{x} \leq \frac{k}{\sigma_L} \boldsymbol{x}^T \boldsymbol{x} + \frac{1}{\sigma_L} \boldsymbol{x}^T \boldsymbol{x} = \frac{k+1}{\sigma_L} \boldsymbol{x}^T \boldsymbol{x}$

In mL-BFGS, $k$ is at most $M$, which is the length of history vector, therefore $\Xi = (M+1)\frac{1}{\sigma_L}$.

In summary, $\frac{1}{\sigma_H} I \preceq \hat{H}_k \preceq (M+1)\frac{1}{\sigma_L} I$ $\qquad\qquad\square$

**Lemma 5.** *Assume AS 2 holds, then loss function $\mathcal{L}(\boldsymbol{\theta})$ is at least $\Lambda$-smooth. At iteration $t$ with mini-batch input $\mathcal{S}_t$, where each sample is randomly sampled from $\mathcal{X}$ with replacement, gradient $\nabla\mathcal{L}(\boldsymbol{\theta}_t; \mathcal{S}_t)$ satisfies*

$$E_{\mathcal{S}_t}[\|\nabla\mathcal{L}(\boldsymbol{\theta}_t; \mathcal{S}_t)\|^2] \leq 2\Lambda \cdot \mathcal{L}(\boldsymbol{\theta}_t).$$

*Proof.* **Smoothness of $\mathcal{L}(\boldsymbol{\theta})$.**

Given AS 2 hold, we have $\|\nabla\ell_i(\boldsymbol{\theta}_1) - \nabla\ell_i(\boldsymbol{\theta}_2)\| \leq \Lambda \|\boldsymbol{\theta}_1 - \boldsymbol{\theta}_2\|$.

For $\mathcal{L}$, we have $\nabla\mathcal{L}(\boldsymbol{\theta}_1) - \nabla\mathcal{L}(\boldsymbol{\theta}_2) = \frac{1}{N}\sum_{i=1}^{N} \nabla\ell_i(\boldsymbol{\theta}_1) - \nabla\ell_i(\boldsymbol{\theta}_2)$.

Then,

$$
\begin{aligned}
\|\nabla\mathcal{L}(\boldsymbol{\theta_1}) - \nabla\mathcal{L}(\boldsymbol{\theta_2})\| &= \frac{1}{N}\left\|\sum_{i=1}^{N}\nabla\ell_i(\boldsymbol{\theta_1}) - \nabla\ell_i(\boldsymbol{\theta_2})\right\| \\
&\overset{\text{TI}}{\leq} \frac{1}{N}\sum_{i=1}^{N}\|\nabla\ell_i(\boldsymbol{\theta_1}) - \nabla\ell_i(\boldsymbol{\theta_2})\| \\
&\leq \frac{1}{N}\sum_{i=1}^{N}\Lambda\|\boldsymbol{\theta_1} - \boldsymbol{\theta_2}\| \\
&= \Lambda\|\boldsymbol{\theta_1} - \boldsymbol{\theta_2}\|
\end{aligned}
$$

"TI" indicates triangle inequality. Therefore, $\mathcal{L}$ is at least $\Lambda$-smooth.

**Gradient variance bound.**

$E_{\mathcal{S}_t}[\|\nabla\mathcal{L}(\boldsymbol{\theta_t};\mathcal{S}_t)\|^2] = E_{\mathcal{S}_t}[\langle \frac{1}{b}\nabla\sum_{i=1}^{b}\ell_i(\boldsymbol{\theta_t}), \frac{1}{b}\nabla\sum_{i=1}^{b}\ell_i(\boldsymbol{\theta_t})\rangle]$

Expand summation and regroup,

$E_{\mathcal{S}_t}[\|\nabla\mathcal{L}(\boldsymbol{\theta_t};\mathcal{S}_t)\|^2] = E_{\mathcal{S}_t}[\frac{1}{b^2}\sum_{i=1}^{b}\|\nabla\ell_i(\boldsymbol{\theta_t})\|^2 + \frac{1}{b^2}\sum_{i=1}^{b}\sum_{j=1,\neq i}^{b}\langle\nabla\ell_i(\boldsymbol{\theta_t}), \nabla\ell_j(\boldsymbol{\theta_t})\rangle]$

Take expectation on each sample,

$E_{\mathcal{S}_t}[\|\nabla\mathcal{L}(\boldsymbol{\theta_t};\mathcal{S}_t)\|^2] = \frac{1}{b^2}\sum_{i=1}^{b}E_{\boldsymbol{x}_i}[\|\nabla\ell_i(\boldsymbol{\theta_t})\|^2] + \frac{1}{b^2}\sum_{i=1}^{b}\sum_{j=1,\neq i}^{b}E_{\boldsymbol{x}_i,\boldsymbol{x}_j}[\langle\nabla\ell_i(\boldsymbol{\theta_t}), \nabla\ell_j(\boldsymbol{\theta_t})\rangle]$

Because $\boldsymbol{x}_i$ and $\boldsymbol{x}_j$ are independent, the second part can be simplified as,

$E_{\mathcal{S}_t}[\|\nabla\mathcal{L}(\boldsymbol{\theta_t};\mathcal{S}_t)\|^2] = \frac{1}{b^2}\sum_{i=1}^{b}E_{\boldsymbol{x}_i}[\|\nabla\ell_i(\boldsymbol{\theta_t})\|^2] + \frac{1}{b^2}\sum_{i=1}^{b}\sum_{j=1,\neq i}^{b}\|\nabla\mathcal{L}(\boldsymbol{\theta_t})\|^2$

With further simplification, we get

$E_{\mathcal{S}_t}[\|\nabla\mathcal{L}(\boldsymbol{\theta_t};\mathcal{S}_t)\|^2] = \frac{1}{b^2}\sum_{i=1}^{b}E_{\boldsymbol{x}_i}[\|\nabla\ell_i(\boldsymbol{\theta_t})\|^2] + \frac{b-1}{b}\|\nabla\mathcal{L}(\boldsymbol{\theta_t})\|^2$

Given AS 2, we have $\|\nabla\ell_i(\boldsymbol{\theta_t})\|^2 \leq 2\Lambda\cdot\ell_i(\boldsymbol{\theta_t})$ and $\|\nabla\mathcal{L}(\boldsymbol{\theta_t})\|^2 \leq 2\Lambda\cdot\mathcal{L}(\boldsymbol{\theta_t})$

Therefore,

$$
\begin{aligned}
E_{\mathcal{S}_t}[\|\nabla\mathcal{L}(\boldsymbol{\theta_t};\mathcal{S}_t)\|^2] &\leq \frac{1}{b^2}\sum_{i=1}^{b}E_{\boldsymbol{x}_i}[2\Lambda\ell_i(\boldsymbol{\theta_t})] + \frac{b-1}{b}2\Lambda\mathcal{L}(\boldsymbol{\theta_t}) \\
&= \frac{1}{b}2\Lambda\mathcal{L}(\boldsymbol{\theta_t}) + \frac{b-1}{b}2\Lambda\mathcal{L}(\boldsymbol{\theta_t}) \\
&= 2\Lambda\cdot\mathcal{L}(\boldsymbol{\theta_t})
\end{aligned}
$$

$\square$

### A.3.2 Proof of Theorem 2

*Proof.* Given AS 2 and Lemma 5, $\mathcal{L}(\boldsymbol{\theta_t})$ can be bounded by $\mathcal{L}(\boldsymbol{\theta_t}) \leq \mathcal{L}(\boldsymbol{\theta_{t-1}}) + \nabla\mathcal{L}(\boldsymbol{\theta_{t-1}})^T(\boldsymbol{\theta_t} - \boldsymbol{\theta_{t-1}}) + \frac{\Lambda}{2}\|\boldsymbol{\theta_t} - \boldsymbol{\theta_{t-1}}\|^2$ for $\forall\boldsymbol{\theta_{t-1}}, \boldsymbol{\theta_t}$

In mL-BFGS, $\boldsymbol{\theta}_t = \boldsymbol{\theta}_{t-1} - \eta_{t-1}\hat{H}_k\nabla\mathcal{L}(\boldsymbol{\theta}_{t-1};\mathcal{S}_{t-1})$. Therefore, we can upper bound $\mathcal{L}(\boldsymbol{\theta}_t)$ as:

$$
\begin{aligned}
\mathcal{L}(\boldsymbol{\theta}_t) \leq & \mathcal{L}(\boldsymbol{\theta}_{t-1}) - \eta_{t-1}\cdot\nabla\mathcal{L}(\boldsymbol{\theta}_{t-1})^T\hat{H}_k\nabla\mathcal{L}(\boldsymbol{\theta}_{t-1};\mathcal{S}_{t-1}) \\
& + n_{t-1}^2\frac{\Lambda}{2}\left\|\hat{H}_k\nabla\mathcal{L}(\boldsymbol{\theta}_{t-1};\mathcal{S}_{t-1})\right\|^2 \\
\leq & \mathcal{L}(\boldsymbol{\theta}_{t-1}) - \eta_{t-1}\cdot\nabla\mathcal{L}(\boldsymbol{\theta}_{t-1})^T\hat{H}_k\nabla\mathcal{L}(\boldsymbol{\theta}_{t-1};\mathcal{S}_{t-1}) \\
& + n_{t-1}^2\frac{\Lambda\Xi^2}{2}\left\|\nabla\mathcal{L}(\boldsymbol{\theta}_{t-1};\mathcal{S}_{t-1})\right\|^2
\end{aligned}
$$

Since $\hat{H}_k$ is independent with $\mathcal{S}_{t-1}$, we take expectation w.r.t $\mathcal{S}_{t-1}$ and $\mathcal{S}_t$,

$$
\begin{aligned}
E_{\mathcal{S}_t}[\mathcal{L}(\boldsymbol{\theta}_t)|\mathcal{S}_{t-1}] \leq & \mathcal{L}(\boldsymbol{\theta}_{t-1}) - \eta_{t-1}\cdot\nabla\mathcal{L}(\boldsymbol{\theta}_{t-1})^T\hat{H}_k E_{\mathcal{S}_{t-1}}[\nabla\mathcal{L}(\boldsymbol{\theta}_{t-1};\mathcal{S}_{t-1})] \\
& + n_{t-1}^2\frac{\Lambda\Xi^2}{2}E_{\mathcal{S}_{t-1}}[\|\nabla\mathcal{L}(\boldsymbol{\theta}_{t-1};\mathcal{S}_{t-1})\|^2] \\
= & \mathcal{L}(\boldsymbol{\theta}_{t-1}) - \eta_{t-1}\cdot\nabla\mathcal{L}(\boldsymbol{\theta}_{t-1})^T\hat{H}_k\nabla\mathcal{L}(\boldsymbol{\theta}_{t-1}) \\
& + n_{t-1}^2\frac{\Lambda\Xi^2}{2}E_{\mathcal{S}_{t-1}}[\|\nabla\mathcal{L}(\boldsymbol{\theta}_{t-1};\mathcal{S}_{t-1})\|^2]
\end{aligned}
$$

According AS 3, we have

$$
L(\boldsymbol{\theta}_{t-1}) \leq \tfrac{1}{\lambda}\left\|\nabla\mathcal{L}(\boldsymbol{\theta}_{t-1})\right\|^2
$$

According to Lemma 5, we have

$$
E_{\mathcal{S}_{t-1}}[\|\nabla\mathcal{L}(\boldsymbol{\theta}_{t-1};\mathcal{S}_{t-1})\|^2] \leq 2\Lambda\mathcal{L}(\boldsymbol{\theta}_{t-1})
$$

Therefore, $E_{\mathcal{S}_t}[\mathcal{L}(\boldsymbol{\theta}_t)|\mathcal{S}_{t-1}] \leq \mathcal{L}(\boldsymbol{\theta}_{t-1}) - \eta_{t-1}\lambda\xi\mathcal{L}(\boldsymbol{\theta}_{t-1}) + \eta_{t-1}^2\Lambda^2\Xi^2\mathcal{L}(\boldsymbol{\theta}_{t-1})$.

After simply regrouping, we can get

$$
E_{\mathcal{S}_t}[\mathcal{L}(\boldsymbol{\theta}_t)|\mathcal{S}_{t-1}] \leq (1 - \eta_{t-1}\lambda\xi + \eta_{t-1}^2\Lambda^2\Xi^2)[\mathcal{L}(\boldsymbol{\theta}_{t-1}]
$$

Apply total expectation rule w.r.t $\mathcal{S}_t$, we have

$$
E_{\mathcal{S}_t}[\mathcal{L}(\boldsymbol{\theta}_t)] \leq (1 - \eta_{t-1}\lambda\xi + \eta_{t-1}^2\Lambda^2\Xi^2)E_{\mathcal{S}_{t-1}}[\mathcal{L}(\boldsymbol{\theta}_{t-1})]
$$

$\square$

## A.4 Hyperparameter Settings

### A.4.1 CIFAR-10/CIFAR-100

For ResNet-18 and DeiT, detailed settings are shown in Table 5. The batch size is set to be 256 for all optimizers. For the learning rate, we use a cosine annealing scheduling strategy with a minimum learning rate of 0.0001. Maximum epoch is 150 for ResNet-18 and 50 for DeiT.

### A.4.2 ImageNet

For ResNet50, detailed settings are shown in Table 6. The batch size is set to be 512 for all optimizers. For the learning rate, we used a cosine annealing scheduling strategy with a minimum learning rate of 0.0001. Maximum epoch is 100.

Table 5: Hyperparameters for SGD, Adam, mL-BFGS on CIFAR-10/CIFAR-100

| Optimizer | $b$ | lr | momentum | $wd$ | $\tau_0$ | $\beta$ | $T$ | $M$ |
|-----------|-----|------|-----------|------|----------|---------|-----|-----|
| SGD | 256 | 0.1 | 0.9 | 1e-4 | - | - | - | - |
| Adam | 256 | 0.01 | 0.9/0.999 | 1e-2 | - | - | - | - |
| mL-BFGS | 256 | 0.1 | 0.9 | 2e-4 | 0.99 | 0.999 | 50 | 10 |

$\beta$: momentum for the Hessian; $\tau_0$: initial damping; $T$: frequency for updating the Hessian; $M$: length of history vector $(\boldsymbol{s}, \boldsymbol{y})$

Table 6: Hyperparameters for SGD, Adam, KFAC and mL-BFGS of ResNet50 on ImageNet

| Optimizer | $b$ | lr | momentum | $wd$ | $\tau_0$ | $\beta$ | $T$ | $M$ |
|-----------|-----|------|-----------|------|----------|---------|-----|-----|
| SGD | 512 | 0.1 | 0.9 | 1e-4 | - | - | - | - |
| Adam | 512 | 0.07 | 0.9/0.999 | 3e-2 | - | - | - | - |
| KFAC | 512 | 0.06 | 0.9 | 3e-2 | - | - | 50 | - |
| mL-BFGS | 256 | 0.1 | 0.9 | 2e-4 | 0.99 | 0.999 | 50 | 10 |

$\beta$: momentum for the Hessian; $\tau_0$: initial damping; $T$: frequency for updating the Hessian; $M$: length of history vector $(\boldsymbol{s}, \boldsymbol{y})$

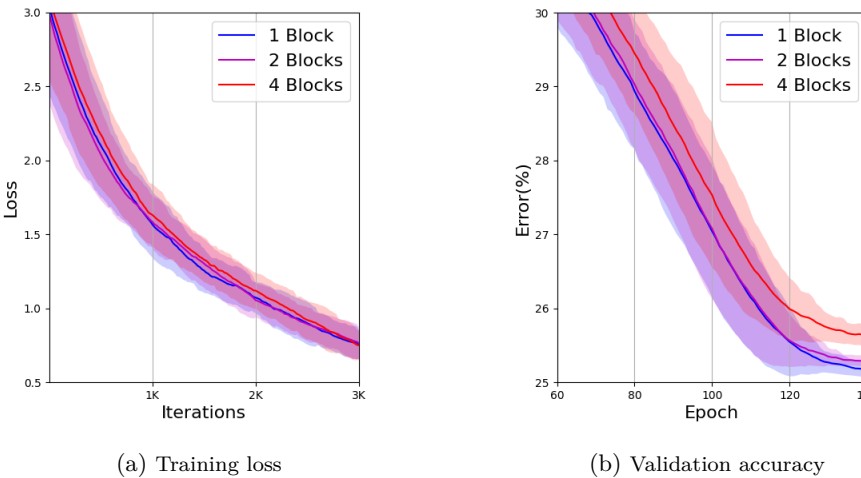

(a) Training loss               (b) Validation accuracy

Figure 6: Training loss and validation accuracy of using mL-BFGS on ResNet-18/CIFAR-100 with different block sizes.

## A.5   Ablation Study: Impact of Granularity of Block-wise Approximation

We study the impact of granularity of block-wise approximation in this section. We train ResNet-18 on CIFAR-100 using mL-BFGS with a different number of Hessian blocks: 1, 2, 4 blocks. mL-BFGS with one block approximates the whole Hessian matrix, while mL-BFGS with two blocks approximates two diagonal Hessian blocks with each one consisting of 4 *resblocks* in ResNet-18. At last, mL-BFGS with four blocks is the default setting in our CIFAR-100 experiment, where each block consists of 2 *resblocks*. As shown in Figure 6, as we increase the size of the blocks, mL-BFGS converges faster and achieve better generalization performance. This observation aligns with the argument that mL-BFGS can estimate more accurate curvature information by approximating a large Hessian block, improving training performance.

## A.6   Justification of PL-condition in Real Neural Networks

Figure 7 shows gradient norm and loss when training ResNet-18 on CIFAR-100. We can observe that with an appropriate parameter $\lambda$, the PL-condition can be easily satisfied.

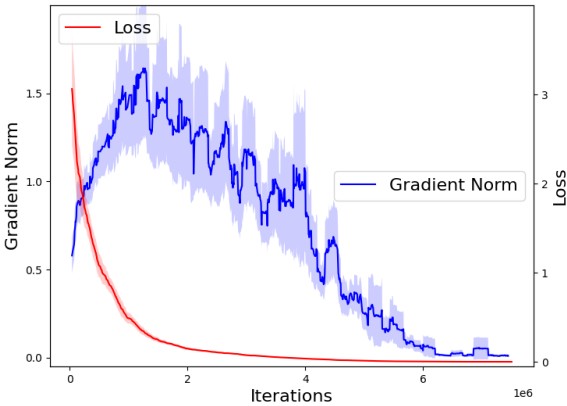

Figure 7: The relation of gradient norm and loss during training (ResNet-18 on CIFAR-100).

## A.7 Other Second-order Methods on CIFAR-10/100

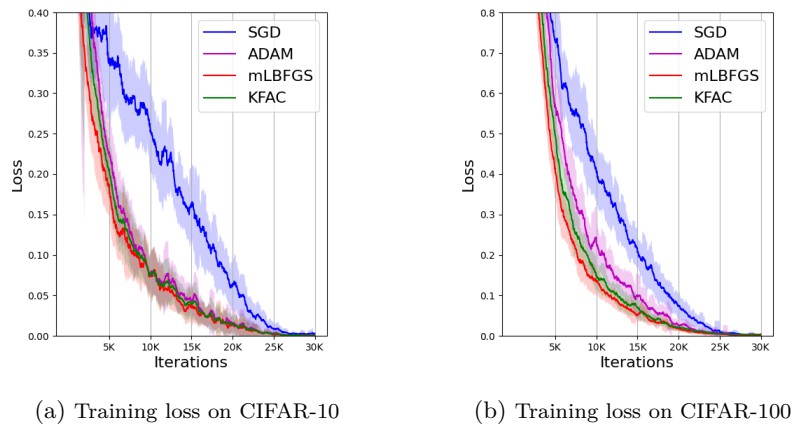

(a) Training loss on CIFAR-10      (b) Training loss on CIFAR-100

Figure 8: Training loss using mL-BFGS, KFAC, Adam and SGD on CIFAR-10/100.

Figure 8 show training loss of using mL-BFGS, KFAC, Adam, and SGD on CIFAR-10/100. Compared to the second-order method KFAC, mL-BFGS converges faster as it can estimate more accurate curvature information by approximating large Hessian blocks. On the validation dataset, mL-BFGS achieves higher accuracy than KFAC (KFAC on CIFAR-10: $93.1 \pm 0.1$, CIFAR-100: $73.2 \pm 0.13$).

