# OpenReview forum: "mL-BFGS: A Momentum-based L-BFGS for Distributed Large-scale Neural Network Optimization"
_TMLR — Accepted by TMLR_

### Review · Reviewer_CWvY · 2023-04-16

**Summary Of Contributions:**

The paper proposes a new second-order optimization method for improving training stability and reducing the computation and memory costs. Specifically, based on L-BFGS, the authors introduce momentum and damping to reduce stochastic noise and stabilize Hessian estimation. They further introduce a block-wise approximation of Hessian to improve memory efficiency in distributed training. Experiments on CIFAR and ImageNet demonstrate the effectiveness of the proposed method.

**Audience:**

Yes

**Broader Impact Concerns:**

No broader impact concerns.

**Claims And Evidence:**

Yes

**Requested Changes:**

-	When comparing to the standard L-BFGS, the authors compare convergence lower bounds (Theorem 1). Can the authors explain why considering lower bounds rather than upper bounds, as upper bounds seem more commonly used in the literature?

-	It is unclear to me how Theorem 2 shows the effect of momentum in the Hessian and block-diagonal approximation.

-	In the CIFAR experiments, it will be better to add some second-order baselines. Besides, since the proposed method is a variant of L-BFGS, it seems reasonable to add it as a baseline.


**Strengths And Weaknesses:**

Strengths

-	The motivation is clear.

-	The proposed method appears to be technically sound

-	The paper is well-written and easy to follow

-	The paper provides both theoretical and empirical analysis

Weaknesses

-	The meaning of the theoretical results is a bit unclear.

-	Missing closely related baselines in experiments.

---

> ### Author Response · Authors · 2023-05-13
> **Thanks and Response**
>
> We thank for the reviewer's time and effort in reviewing our work. Please see our response below.
>
> `1. Can the authors explain why considering lower bounds rather than upper bounds, as upper bounds seem more commonly used in the literature?`
>
> We are not showing the convergence of L-BFGS in Theorem 1. Instead, we want to show that, when using noisy stochastic gradients to estimate the Hessian in L-BFGS, the *best achievable* loss by L-BFGS is lower-bounded by the variance in the noisy gradients. If the stochastic gradients have large variance, the  *best achievable* loss will also be higher, no matter what hyperparameters are used during training. Theorem 1 provides a crucial insight into why reducing noise in gradient when estimating the Hessian using L-BFGS. Furthermore, it also lays the foundations for the effectiveness of momentum-based L-BFGS.
>
> `2. It is unclear how Theorem 2 shows the effect of momentum in the Hessian and block-diagonal approximation`
>
> First, Theorem 2 shows the convergence of mL-BFGS with damping involved. The convergence rate $\alpha_{t}$ is mainly affected by damping parameters $\sigma_L, \sigma_H$.
>
> For the effect of momentum, we first theoretically show in Lemma 1 and Theorem 1 that it reduces stochastic noise in the Hessian approximation and can achieve a lower loss in a non-damping L-BFGS compared to standard L-BFGS. Though not captured in the convergence analysis in Theorem 2, those benefits are further reflected in real-model training. As shown in Sec 5.3, Without momentum, model training is severely affected by stochastic noise in gradients, even with the Hessian damping introduced. L-BFGS with momentum significantly reduces the effects of stochastic noise and smoothing the training curve.
>
> As for the effect of block-diagonal approximation, Lemma 3 shows how the full Hessian approximation is affected by the Hessian of each diagonal block. Specifically, the lower and upper bound of the full Hessian is decided by the smallest lower bound and largest upper bound among the Hessian blocks.
>
> Please let us know if any further clarifications are needed.
>
> `3. In the CIFAR experiments, it will be better to add some second-order baselines.`
>
> **We have provided more second-order baselines in CIFAR experiments in Appendix A.7**. Compared to the second-order method KFAC, mL-BFGS gives faster convergence as it can estimate more accurate curvature information by approximating large Hessian blocks. As we showed in the main paper, the key contribution of mL-BFGS is to estimate the Hessian with marginal costs, therefore leading to a much better wall-clock convergence performance. Such a benefit is more striking in large-scale model training. Therefore, we use ResNet-50 on ImageNet as the main study case to compare their wall-clock performance.
>
> `3.1. Besides, since the proposed method is a variant of L-BFGS, it seems reasonable to add it as a baseline`
>
> We also ran standard L-BFGS in model training. However, as we showed in Sec 5.3,  training with standard L-BFGS is very easy to diverge. Therefore, we did not include it in our main experiments. Other L-BFGS variants require prohibitively high complexities (See Table 2), making applying them in real model training difficult and time-consuming.

---

### Review · Reviewer_fNCc · 2023-04-17

**Summary Of Contributions:**

The authors propose a momentum-based L-BFGS algorithm, mL-BFGS,  for large scale distributed optimization of deep neural networks.   There are two major algorithmic contributions. One is introducing momentum to the stochastic L-BFGS and also with damping.  The other is model parallelization across learners in the distributed setting to reduce the computational and memory costs, which amounts to a block-wise Hessian approximation. mL-BFGS is a light weight second-order stochastic optimizer.  It takes into account the curvature information of the optimization landscape and therefore is supposed to be faster in convergence.  The authors give the convergence rate, analyze its complexity and show its effectiveness by comparing to existing techniques on various datasets.

**Audience:**

Yes

**Broader Impact Concerns:**

I don't have concerns on the ethical implications of the work.

**Claims And Evidence:**

Yes

**Requested Changes:**

Please refer to my questions and concerns above.

**Strengths And Weaknesses:**

1. First of all, there is no explicit mention of what "m" stands for in mL-BFGS to begin with.  I think it is for "momentum" but am not sure.  It would be helpful to make it clear.

2. I wonder what the convergence rate is without the assumption of the PL-condition.  Usually the convergence rate of SGD or its variants is given as  $O(\frac{1}{\sqrt{T}})$ under the assumptions on Lipschitz continuous gradients, its unbiasedness and bounded variance for non-convex objective functions.  It would be helpful to show the convergence rate of mL-BFGS under the same assumptions. PL-condition seems a bit strong to me.

3. In Eq.5, do $\theta$ and $g$ have to share the same momentum parameters $\beta$ ?

4. The block-wise approximation of the Hessian matrix using model parallelization is a very popular technique in the ML community and other Quasi-Newton methods can also readily use it.  I am not sure this can be considered a unique part of the design of mL-BFGS.  Also,  it would be helpful to analyze the impact of granularity of block-wise approximation to the performance.

5. I am bit confused by the loss curves in Fig 3 and accuracies in Tab. 3 for CIFAR-10/100.  Do they show that in order to get the best accuracy, mL-BFGS has to run as many iterations as SGD and Adam?  If the required numbers of iterations are the same for the best accuracy, then what is the advantage of using mL-BFGS since it requires more computations and memory?  I would like to see the iterations used to get the accuracies reported in Tab. 3 under SGD, Adam and mL-BFGS.

6.  I have a concern that mL-BFGS may not be large-batch friendly.  Large scale distributed optimization with large number of learners (say, 64 or 128) relies on large batch size. Given the extra memory used by mL-BFGS, the local affordable batch size on each learner may be smaller than SGD, if not much smaller.


7.  I would suggest the authors also verify the effectiveness of  mL-BFGS on models using RNN/LSTM architectures.

---

> ### Author Response · Authors · 2023-05-13
> **Thanks and Response**
>
> We thank for reviewer's time and effort in reviewing our work. Please see our response below.
>
> `1. What mL-BFGS stands for?`
>
> Yes, "m" stands for "momentum".  We have clarified that in the updated manuscript.
>
> `2. Convergence rate without PL-condition.`
>
> First, the PL-condition is not a strong assumption. In fact, in real neural networks, the PL-condition can be easily satisfied. **In Appendix A.6, we studied the relation between the gradient norm and loss**. We can see that with an appropriate parameter $\lambda$, the PL-condition in *AS 3* is easily attained.
>
> Besides, the central part of the convergence analysis is to show the Hessian approximation using mL-BFGS is bounded (See A.3). With the bounded Hessian, we follow a similar procedure as SGD to derive convergence of mL-BFGS. Therefore, in terms of theoretical convergence, mL-BFGS gives a rate with the same order as SGD under the same assumptions.
>
> `3. In Eq.5, do` $\theta$ and $g$ `share the same momentum parameter` $\beta$.
>
> Yes, $\theta$ and $g$ share the same momentum parameter $\beta$.
>
> `4.1 The block-wise approximation of the Hessian matrix is a very popular technique in the ML community and other Quasi-Newton methods can also readily use it. I am not sure this can be considered a unique part of the design of mL-BFGS.`
>
> mL-BFGS provides a more flexible block-wise configuration. As we showed in Sec 3.2, mL-BFGS allows each diagonal Hessian block to cover *an arbitrary number of layers* and guarantees the entire Hessian approximation is well bounded. Methods such as KFAC can only make a layer-wise approximation with each block covering *one layer*. Owing to the flexible diagonal block approximation, mL-BFGS captures more accurate curvature information than KFAC. Moreover, such a flexible approximation allows a balanced workload of computing the Hessian and gradient conditioning on different nodes. If resource constraints vary among nodes, mL-BFGS allows each node to approximate a Hessian block with a different number of layers.
>
> `4.2 It would be helpful to analyse the impact of granularity of block-wise approximation to the performance.`
>
> Thanks for the suggestion. We have conducted an ablation study on the impact of granularity of block-wise approximation, and **include the result in Appendix A.5**. The study shows that training performance is improved when approximating the Hessian with large block sizes. Such an observation aligns with the argument that mL-BFGS can estimate more accurate curvature information by approximating a large Hessian block compared to other second-order methods.
>
> `5.1 Do they show that in order to get the best accuracy, mL-BFGS has to run as many iterations as SGD and Adam?.`
>
> Yes, mL-BFGS needs almost the same number of iterations as SGD and Adam to achieve the best accuracy.
>
> `5.2 If the required numbers of iterations are the same for the best accuracy, then what is the advantage of using mL-BFGS since it requires more computations and memory?`
>
> While mL-BFGS needs almost the same number of iterations to achieve the best accuracy, it does not mean mL-BFGS does not have advantages. In Figure 3 and 4, we observe that mL-BFGS achieve much faster early-stage convergence. Late-stage convergence is mainly affected by the decayed learning rate no matter which optimizer is used. The faster early-stage convergence in mL-BFGS matters when training a model is time-consuming and a near-optimal accuracy is accepted. Specially, as shown in Figure 4.c, if there are not enough resources and time to train a large-scale model, mL-BFGS offers a better option to early-stop training with much higher accuracy.
>
> `6. mL-BFGS may not be large-batch friendly.`
>
> Thanks for bringing up this discussion. mL-BFGS is large-batch friendly. While mL-BFGS uses additional memory to store second-order information, the memory footprint decreases with the number of nodes (See Table 1). For a large model trained with large batch sizes, each node can choose to approximate smaller diagonal blocks as the number of nodes increases. Therefore, per-node memory on the second-order information becomes more and more marginal compared to memory on forward and backward passes (See Table 1).
>
> `7. Verify the effectiveness of mL-BFGS on models using RNN/LSTM architecture.`
>
> It is crucial to evaluate mL-BFGS on different model architectures. The reason for choosing CNN and Transformer models is that they are widely used in current vision and language tasks, especially for some large-scale problems. Transformer models have replaced RNN/LSTM architectures and perform much better in many vision and language tasks. Therefore, evaluations on CNNs and Transformers provide sufficient evidence of mL-BFGS's performance compared to other baselines.

---

### Review · Reviewer_xAnx · 2023-05-08

**Summary Of Contributions:**

This paper proposed mL-BFGS, a distributed stochastic Quasi-Newton for large-scale model training. mL-BFGS contains two main designs: (1) momentum for Hessian approximation, which reduces the stochastic noise; (2) a block-wise approximation method that allows distributed storage for model layers and thus enables efficient distributed training. The authors give convergence proof and complexity analysis on the memory. Empirically, comparison between mL-BFGS and other algorithms (SGD, Adam, etc.) is done on CIFAR and ImageNet.

**Audience:**

Yes

**Broader Impact Concerns:**

There is no foreseeable ethical implication.

**Claims And Evidence:**

Yes

**Requested Changes:**

- Please consider clarifying Weakness 1 and 2.
- Please consider adding some text to answer Weakness 3.

**Strengths And Weaknesses:**

Strengths

- The proposed method is novel and new (to my knowledge). And it is an important step for the traditional L-BFGS algorithm to scale.
- The theoretical contribution is sufficient, the authors provide analysis to both convergence analysis and memory usage.

Weaknesses

- In the original [ResNet paper](https://arxiv.org/abs/1512.03385), momentum-based SGD with decaying learning rate is used, in order to reach target accuracy for both CIFAR and ImageNet. I'm not sure if the current learning rate schedule (cosine annealing scheduling strategy) without momentum for SGD is comparable to that, and why authors choose this schedule.
- The experiments are small-scale. Despite DeiT-Tiny is a transformer model, its size is much smaller than ResNet-18, is there a reason why the authors choose DeiT-Tiny instead of DeiT-Base or DeiT-Small? Is it because of the cost for memory consumption?
- Last but not least, I wonder if there's any particular reason second-order optimizer should be preferred than the first-order optimizer for model training, especially distributed training in general?

---

> ### Author Response · Authors · 2023-05-13
> **Thanks and Response**
>
> We thank the reviewer for acknowledging the novelty and contributions in the paper. Please see our response below.
>
> `1. I am not sure if the current learning rate schedule (cosine annealing) without momentum for SGD is comparable to that, and why authors choose this schedule.`
>
> First, as we provide in A.4 (Hyperparameter Settings), the baseline training using SGD has momentum (See Table 5 and Table 6). We adopt standard hyperparameter settings for SGD on CIFAR10/100 and ImageNet datasets.
> For the choice of learning rate scheduler, we tested both step decay (used in the original ResNet paper) and cosine annealing decay. The cosine annealing decay gives higher 1-crop test accuracy on CIFAR10/CIFAR100 and ImageNet, which aligns with the original paper proposing a cosine annealing scheduler [1]. Further, with a cosine annealing scheduler, comparing convergence speed for different methods is more straightforward. Therefore, with these benefits, we use the cosine annealing scheduler in our main experiments.
>
> `2. The experiments are small-scale. Is there a reason why the authors choose DeiT-Tiny instead of DeiT-Base or DeiT-Small? Is it because of the cost for memory consumption?`
>
> First, we conduct both small- and large-scale experiments. In small-scale experiments, we test various models in CIFAR-10/100. For choosing DeiT-Tiny, we want to show that mL-BFGS consistently performs better on different model architectures (CNNs, Transformers).  Deit-Tiny (pre-trained on ImageNet) has very high accuracy on CIFAR-100, so it is a suitable transformer model to test convergence performance and generalization performance for different optimizers.
>
> As for large-scale experiments, we use ResNet-50 on ImageNet as a study case to show mL-BFGS not only achieves faster per-epoch convergence, but also conveys such convergence in wall-clock time. The large-scale experiments on ResNet-50 provide strong evidence that mL-BFGS reduce the complexities of second-order methods but still preserve the convergence speed.
> As large-scale transformers need much more time to converge, as well as higher memory requirements, we will consider evaluating mL-BFGS on them in the future.
>
> `3. I wonder if there's any particular reason second-order optimizer should be preferred than the first-order optimizer for model training, especially distributed training in general`
>
> Thanks for bringing up the discussion. An optimizer is preferred if it converges faster and has good generalization performance. That is the motivation for exploring second-order methods in distributed large-scale model training. Second-order optimizers can potentially deliver faster per-iteration convergence than first-order optimizers, especially when training large-scale models and large datasets (e.g., ResNet-50 on ImageNet). The key problem is to maintain such fast convergence speed in real time. mL-BFGS achieves this goal and trains a model much faster than first-order methods, even in distributed systems. In many applications, faster convergence means less time for each training round. Therefore, we will spend less time obtaining a good-to-use model. Furthermore, given a computation and time budget, we can have more training rounds to find the optimal hyperparameters. In summary, **fast real-time convergence** is the key reason for using second-order methods.
>
> *Reference*
>
> [1] Loshchilov, I. and Hutter, F., *SGDR: Stochastic Gradient Descent with Warm Restarts*. In ICLR 2017.

---

### Author Response · Authors · 2023-05-13
**Summary of changes in the updated manuscript**

Dear reviewers

Thanks for your time in reviewing our work. We have updated the manuscript based on the reviewers' suggestions. The main changes are summarized below.

1. We add an ablation study on the **impact of granularity of block-wise approximation** in A.5 (Reviewer *fNCc*). We observe that as mL-BFGS approximate large Hessian blocks, convergence, and generalization performance become better.

2. We provide a **justification of the PL-condition** in real neural networks in A.6, which shows the PL-condition can be met in real neural nets (Reviewer *fNCc*).

3. We add more **second-order experiments** on CIFAR-10/100 in A.7 (Reviewer *CWvY*).

Please kindly let us know if you have other questions.

Best, Authors

---

### Public Comment · ~Apostolos_Avranas1 · 2024-11-18
**Clarifications for the algorithm**

Dear authors,

We are trying to implement and reproduce your results. Could you share your code ?

In case you cannot share it, we would need some additional information. What are the values of $\sigma_H, \sigma_L$ ? Also, the aglorithm 2 is a L-BFGS which should output $\Delta\theta_t$ and not $g_t$. Is it a typo or Algorithm 2 outputs $g_t$ that is multiplied ("again") with $\hat{H}_k^i$ in line 10 of Algorithm 1 ?

Finally, it seems more reasonable in Algorithm 1 the lines 15-18 to be before lines 10-11, because every $T$ iterations it is possible to first update the approximated inverse hessian $\hat{H}_k^i$ and then to find the $\Delta\theta_t$ using the already updated  $\hat{H}_k^i$ . Instead you choose to use the updated  $\hat{H}_k^i$  only for the next iterations. Is there any particular reason for this choice ?

Thank you

---

> ### Author Response · Authors · 2024-11-24
> **Clarification**
>
> **Code**: https://github.com/yuehniu/mL-BFGS
>
> For your questions:
>
> - For Alg 2, the output is $g_t$, a pre-conditioned gradients with the approximated Hessian.
>
> - For Alg 1, I think either way is okay. The Hessian gets updated for every $T$ iterations. As a result, no matter lines 15-18 are before or after lines 10-11, the Hessian approximation is a little bit stale.
>
> Thanks for your interest. Let me know if anything else is unclear.

---

> > ### Public Comment · ~Apostolos_Avranas1 · 2024-12-03
> > **Inconsistencies paper with code**
> >
> > Thank you for sharing the code. It is well written and helped us understand how you implement the algorithm. Unfortunately we see that there is a mismatch between the code and the paper. Starting from the algorithm in "/main/opt/slim.py" which is a simpler version of "main/opt/slimblock.py", we understand that it does the following steps:
> >
> > ### Algorithm 1: mL-BFGS Algorithm $(T, M, \theta, \beta, wd, \mu)$
> >
> > 1. **Initialize** $\theta_0$, $\mathcal{M}^i_{\theta_0} = \theta^i_0$, $\mathcal{M}^i_{g_0} = g^i_0$, $\mathbf{b_0}=0$:
> > 2. **For**   $t = 1,\dots, $max_iter:
> >    1. Randomly choose mini-batch input $X_t \in \mathcal{X}$
> >    2. Perform model forward and backward, compute gradients $g_t$ given $X_t$
> >    3. **For each parameter block $i$**:
> >       A: $\mathcal{M}^i_{\theta_t} = \beta \cdot \mathcal{M}^i_{\theta_{t-1}} + (1 - \beta) \cdot \theta^i_t$
> >       B: $\mathcal{M}^i_{g_t} = \beta \cdot \mathcal{M}^i_{g_{t-1}} + (1 - \beta) \cdot g^i_t$
> >       C: **If** $t \% T == 0$ and $t > T$:
> >          - $k = k + 1$
> >          - $s^i_k = \mathcal{M}^i_{\theta_t} - \mathcal{M}^i_{\theta_{t-T}}$
> >          - $y^i_k = \mathcal{M}^i_{g_t} - \mathcal{M}^i_{g_{t-T}}$
> >          - **Apply damping:** $\hat{y}^i_k = \tau \cdot y^i_k + (1 - \tau) \cdot s^i_k$  with $(\sigma_L, \sigma_H)=(0.01, 1.0)$
> >
> >       D: **If** $t \leq 2T$:
> >          - **Warmup with SGD:** $\theta^i_{t+1} = \theta^i_t - \eta_t \cdot g^i_t$
> >
> >       E: **Else**:
> >       - $g^i_t \leftarrow g^i_t + wd*\theta_t^i$ ($wd$ = weight decay)
> >       - Use Alg. 2 (L-bfgs) to transform  $g^i_t$ into $\hat{g}^i_t$
> >         *("pre-conditioning" the gradients)*
> >       - Scale down $\hat{g}^i_t$ if necessary so as the norm not to exceed the value args.grad_clip
> >       -  $\mathbf{b_t} \leftarrow \mu \cdot \mathbf{b_{t-1}} +\hat{g}^i_t  $
> >       - $\theta^i_{t+1} = \theta^i_t - \eta_t \cdot \mathbf{b_t}$
> >    4. **End For**
> >
> > 5. **End For**
> >
> > Some of the minor inconsistencies are that you use weight decay, you scale down the gradients and the step 2.3.C comes before the 2.3.E (which was the comment I had initially done). We find those differences minor but there are two major conserns:
> > *  The code of the file "main/opt/slimblock.py" (which is the one you finally use) has an extra "momentum-like" step with the respect to the above "/main/opt/slim.py"  (see lines " dp_flatten.mul_( 1-mmm ).add_( self.hist_dp[ bk ][-1], alpha=mmm )" and "dg_flatten.mul_( 1-mmm ).add_( self.hist_dg[ bk ][-1], alpha=mmm )"  ). This is applied on  the pair $(s^i_k,y^i_k)$ step 2.3.C just before the dampening. This seems to be a very important step which should appear in the paper and also justified.
> >  * Most importantly, the updates of the parameters which happen in step 2.3.E are the same to the standard SGD with momentum (note that in the code also momentum equal to $\mu=0.9$ ). The main difference from SGD is that instead of using immediately the gradients   $g^i_t$, first the Algorithm 2 transforms them to retrieve the "preconditioned" $\hat{g}^i_t$. The step 10 (of Alg. 1 from the paper) that multiplies the (inverse) Hessian with the preconditioned gradients doesn't happen in the code. This could be quite problematic, because in case  $g^i_t \approx \hat{g}^i_t$ the implemented code is approximately a SGD with momentum and not a Quasi-Newton method.
> >
> > Thank you again for sharing the code and please share your response to our concerns.

---

> > > ### Author Response · Authors · 2024-12-06
> > > **Response**
> > >
> > > Hi,
> > >
> > > 1. The extra momentum is an additional factor introduced when we were trying to further improve the implementation. It further helps reduce variance in the gradients. Thanks for pointing it out.
> > >
> > > 2. One key difference between mL-BFGS and SGD is that gradients are pre-conditioned in mL-BFGS before performing updating (See lines 380-382 in slimblock.py).  For
> > >
> > > > The step 10 (of Alg. 1 from the paper) that multiplies the (inverse) Hessian with the preconditioned gradients doesn't happen in the code. This could be quite problematic, because in case $g_t^i \approx \hat{g}_t^i $, the implemented code is approximately a SGD with momentum and not a Quasi-Newton method.
> > >
> > > I didn't quite get it. In mL-BFGS, preconditioned gradients are the gradients that are multiplied by the inverse Hessian. I didn't get it why discuss the case, $g_t^i \approx \hat{g}_t^i $ ?

---

> > > > ### Public Comment · ~Apostolos_Avranas1 · 2024-12-08
> > > > **Response**
> > > >
> > > > Sorry for not articulating well my thoughts. The algorithm you implemented is the same as SGD with the difference that instead of using immediately the gradients   $g^i_t$ you "pre-condition" them into $\hat{g}^i_t$. I can see two problems:
> > > >   1. Practical problem: To transform the gradients $g^i_t$ into $\hat{g}^i_t$ you add some complicated steps. All this additional complexity should be justified by an improvement in the performance compared to the standard SGD using  $g^i_t$. According to your results SGD works better with    $g^i_t$ so there is no need for the extra steps.
> > > >   2. Conceptual: I know that the term "second-order" methods and "quasi-newton" are very broad. Nonetheless, a Newton method  should somehow use second-order information (i.e. curvature), find the optimal step $\Delta \theta$ to reach the (approximated) stationary point and maybe then scale it down before applying it (so as to satisfy wolfe conditions). But your proposal doesn't find a step but a direction  $\hat{g}^i_t$. Then step taken is not even in that direction since it is plugged into the momentum $\mathbf{b}_t$. Therefore I do not see how you actually you use the second order information.
> > > >
> > > >  I think that in order to prove that you do something different than SGD you need to show that the direction pointed by $g_t$ is (on average) different than $\hat{g}_t$. My fear is that the four momentum you added (on $\theta_t, g_t$ and the two not mentioned on the paper on $s_t, y_t$) push the algorithm 2 to approximate the inverse hessian as an identity matrix (maybe multiplied by a scalar $\gamma$) and so the output is $g_t\approx \hat{g}_t$. This unfortunately would mean that there is not much difference to SGD (a first order method).

---

### Decision · Action_Editors · 2023-07-17

**Recommendation:** Accept with minor revision

**Comment:**

The paper is reviewed by three expert reviewers and all three agree that the paper can be accepted. That said, all three reviewers are "leaning accept" as such, this makes it hard to accept the paper as is. I kindly ask the authors to go over all reviewer concerns one final time and address any remaining issues that remains. After this, I believe the paper will be in great shape for publication at TMLR.

**Audience:**


This is a very well written optimization paper which has applications to machine learning, as clearly demonstrated by the authors through numerical studies. It certainly has value to the TMLR community.

**Claims And Evidence:**

This paper suggests a novel approach to enhance training stability and minimize computation and memory requirements within a second-order optimization framework.

The theoretical claims have been proven rigorously and the experimental setup is explained clearly. Authors also added additional experiments to address some reviewer concerns.

The paper is well written and the proposed mL-BFGS seems to have its value to the TMLR community.

---

> ### Author Response · Authors · 2023-07-25
> **The camera ready is uploaded**
>
> Dear Action Editor,
>
> Thank you for managing the review process, as well as your insightful feedback on the work. We have addressed the reviewers' concerns and uploaded the camera-ready version. Please let us know if any issues need to be resolved.
>
> We will upload the codebase and a short video presentation soon.
>
> Best,
> The Authors